# CRL4^DCAF12 regulation of MCMBP ensures optimal licensing of DNA replication

Anoop Kumar Yadav [1,2,5], Alikhan Abdirov[3,5], Katarina Ondruskova[3], Simran Negi [1,2], Kristina Jamrichova[3], Karolina Kolarova[3,4], Nikol Dibus [3], Jana Krejci [1], Hana Polasek-Sedlackova [1] ✉ & Lukas Cermak [3] ✉

The minichromosome maintenance (MCM2-7) protein complexes are central drivers of genome duplication. Distinct protein pools, parental and nascent MCMs, and their precise equilibrium are essential to sustain error-free DNA replication. However, the mechanism responsible for generating these pools and maintaining their equilibrium remains largely unexplored. Here, we identified CRL4^DCAF12 as a factor controlling the assembly of nascent MCM complexes. During MCM biogenesis, MCMBP facilitates the assembly and transport of newly synthesized MCM3-7 subcomplexes into the nucleus. Once in the nucleus, the MCM2 subunit must be incorporated into the MCM3-7 subcomplex, while MCMBP needs to be removed. CRL4^DCAF12 facilitates the degradation of MCMBP and thereby regulates the assembly of MCM2-7 complexes. The absence of CRL4^DCAF12 adversely affects the level of chromatin-bound nascent MCMs, resulting in accelerated replication forks and replication stress. Collectively, our findings uncovered the molecular mechanism underlying nascent MCM production essential to counteract genome instability.

Accurate and complete genome duplication is a vital process that relies on error-free regulation of DNA replication. In eukaryotic cells, DNA replication is initiated in the G1 phase by origin licensing, during which inactive MCM2-7 protein complexes are loaded on replication origins[1]. Subsequently, during the S phase, a small fraction of inactive MCM complexes is converted to active replicative CMG (CDC45-MCM2-7-GINS) helicases, around which the entire replisome responsible for genome duplication is assembled. Although the majority of MCM complexes remain inactive throughout the DNA replication program, their precise regulation and maintenance during the cell cycle are critical for cellular fitness and survival. Our recent work provided mechanistic insights into this MCM paradox[2]. We found that two distinct MCM protein variants sustain optimal origin licensing and error-free genome duplication in daughter cells: (1) parental MCMs, which represent a protein pool already involved in the DNA replication

process in mother cells; and (2) nascent MCMs, which were newly formed in mother cells, but not used in the replication process. Notably, these two MCM protein forms differ not only in age but also in their engagement during the replication program. While parental MCM complexes are preferentially converted to active replicative CMG helicases, the nascent MCMs remain largely inactive but function as natural replisome pausing sites[3]. Such pausing sites help to regulate the physiological replication fork progression and thereby maintain error-free genome duplication. Therefore, even a minor imbalance in MCM levels can cause replication-induced DNA damage, which, if left unrepaired, can lead to genome instability and severe diseases, including cancer[4–8]. Although numerous regulatory pathways that oversee MCM chromatin binding have been identified in eukaryotic cells[9,10], the apical mechanism responsible for the generation and maintenance of MCM equilibrium across generations of dividing cells

[1]Institute of Biophysics, Czech Academy of Sciences, Brno, Czechia. [2]Department of Experimental Biology, Faculty of Science, Masaryk University, Brno, Czechia. [3]Laboratory of Cancer Biology, Institute of Molecular Genetics, Czech Academy of Sciences, Prague, Czechia. [4]Faculty of Science, Charles University, Prague, Czechia. [5]These authors contributed equally: Anoop Kumar Yadav, Alikhan Abdirov. ✉e-mail: polasek-sedlackova@ibp.cz; lukas.cermak@img.cas.cz

remains largely unexplored. In this work, using a set of screens against cullin (CUL)-RING E3 ubiquitin ligases (CRLs) and their receptors, we identified the DCAF12 substrate receptor as a regulator of the assembly of nascent MCM complexes. During the MCM biogenesis pathway, the MCMBP, a specific chaperone, promotes the assembly and nuclear transport of newly formed MCM3-7 complexes[2,11]. Subsequently, MCMBP needs to be removed as it is not part of pre-replicative complexes loaded on chromatin. We showed that this step is regulated by CRL4[DCAF12], which facilitates the formation of a complete MCM2-7 ring. Inactivation of the MCM biogenesis pathway leads to suboptimal origin licensing, which results in fast replication fork progression and genome instability. Collectively, these findings reveal a cellular mechanism important to sustaining optimal MCM equilibrium and, thus, error-free genome duplication.

## Results

### siRNA-based screening identified CRL4[DCAF12] as a regulator of MCM equilibrium

The ubiquitin-proteasome system (UPS) is a central mechanism through which cells achieve selective protein degradation, necessary to sustain the homeostasis of essential proteins within a cell[12]. Therefore, we aimed to investigate the potential role of UPS in the generation and maintenance of MCM equilibrium. To this end, we treated U2OS cells with MLN4924, an inhibitor of all CRLs, and monitored the level of MCMBP, a specific chaperone for the nascent MCM complexes during their biogenesis[2,11]. Quantitative image-based cytometry (QIBC) revealed a higher protein level of MCMBP upon MLN4924 inhibitor treatment (Supplementary Fig. 1a). Next, we opted for a siRNA-based screening approach to directly target individual cullins (CUL1, CUL2, CUL3, CUL4A/B, and CUL5). CRL4[AMBRA1] has previously been identified as a regulator of D-type cyclins[13,14]; thus, the cyclin D1 marker was utilized as a positive control for siRNA cullin screening (Supplementary Fig. 1b-d). Notably, the depletion of CUL4A/B resulted in a massive stabilization of MCMBP, indicating a putative regulatory role of CRL4A/B on MCMBP (Fig. 1a-c). The C-terminus of CUL4A/B interacts with RBX1 to recruit the E2 enzyme, while the N-terminus binds specific substrate receptors via DNA-damage-binding protein 1 (DDB1). Previous research identified DDB1- and CUL4A/B-associated factors (DCAFs), which mediate substrate protein recognition and its recruitment[15,16]. To determine the substrate receptors of CUL4A/B complexes that are involved in the regulation of MCMBP, we conducted a siRNA screen targeting DCAFs substrate receptors. In this screen, CDT2 (DCAF2) and AMBRA1 (DCAF3) receptors served as positive controls. Aligned with the previous observations[13,14,17,18], depletion of CDT2 and AMBRA1 resulted in higher stabilization of CDT1 and cyclin D1 proteins, respectively (Supplementary Fig. 1e-h). Notably, the depletion of DCAF12 led to elevated MCMBP levels (Fig. 1d-f). To clarify whether this effect was caused by increased MCMBP stabilization due to impaired protein degradation or by elevated mRNA levels, we measured the relative mRNA level of MCMBP. Our findings revealed that DCAF12 depletion did not show any impact on steady-state MCMBP mRNA levels (Supplementary Fig. 1i-j). This suggests that CRL4[DCAF12] functions as a specific regulator of MCMBP, through which it can potentially modulate the cellular MCM equilibrium.

### CRL4[DCAF12] specifically interacts with MCMBP through the C-terminal degron motif

Prior research demonstrated that DCAF12 recognizes and interacts with its protein substrates through their short peptide motifs, termed degrons, located at their C-terminus[19]. To investigate the potential interaction between MCMBP and DCAF12, we first re-analyzed mass spectrometry (MS) data from our affinity purification of StrepII-Flag-tagged DCAF12[20], which was followed by immunoprecipitation of HA-tagged CUL4A (Fig. 1g). To enrich for protein substrates specifically

interacting with CRL4[DCAF12], the resulting interactomes were plotted as the ratio of either of two control CRLs (CRL4A[DCAF4] or CRL1[FBXL6]) to CRL4A[DCAF12] (Fig. 1h, i). This analysis pinpointed MCMBP as one of the top interacting partners of the fully assembled CRL4A[DCAF12] complex (Fig. 1h, i). To further understand whether the association of MCMBP with CRL4A[DCAF12] is mediated directly through DCAF12, we analyzed co-purified proteins using MS immediately after affinity purification with StrepII-Flag-tagged DCAF12. Similarly to the tandem approach, this analysis revealed MCMBP among prominent DCAF12-interacting partners localized in the nucleus (Supplementary Fig. 2a-f). Our extensive MS analysis demonstrated the association of MCMBP with DCAF12 within a fully assembled CRL4 complex. The interaction between MCMBP and DCAF12 was recapitulated by western-blotting-based immunoprecipitation, confirming that the binding between DCAF12 and MCMBP is unique among different CRL1 and CRL4 adapters (Fig. 1j). Next, we sought to explore whether the interaction between MCMBP and DCAF12 is directly mediated through the C-terminal degron motif. Recent studies identified that beyond the canonical twin-glutamic acid -EE* degron motif, CRL4[DCAF12] can recognize protein substrates ending with alternative motifs, such as -EI*, -EM*, -ES*, and -EL* [refs. 20,21]. Indeed, the bioinformatic analysis revealed a putative -EL* degron motif in MCMBP protein conserved across higher eukaryotes (Supplementary Fig. 2g, h), the deletion of which abrogated the interaction between MCMBP and DCAF12, similarly as the deletion of the C-terminal -EL* degron in MOV10, a previously identified DCAF12 substrate (Fig. 1k)[20]. Besides DCAF12, the mammalian proteome contains two paralogs, DCAF12L1 and DCAF12L2, which share approximately 60% amino-acid sequence identity with the canonical DCAF12 protein[22]. Nevertheless, the immunoprecipitation of DCAF12 and its paralogs revealed the interaction only between canonical DCAF12 and MCMBP, suggesting DCAF12 paralogs are not functionally redundant with DCAF12 (Supplementary Fig. 2i). Based on these data, we conclude that CRL4[DCAF12] selectively recognizes MCMBP as a C-terminal degron substrate.

### MCMBP protein levels are tightly regulated by CRL4[DCAF12]-mediated ubiquitination and proteasomal degradation

Multiple siRNA screens (Fig. 1) showed that DCAF12 depletion increased MCMBP protein levels. Similar results from pharmacological inhibition or CUL4A/B knockdown suggest that this effect is due to impaired CRL4-dependent protein degradation. This observation was reproduced with two independent siRNAs targeting different regions of the *DCAF12* gene (Supplementary Figs. 1i and 3a-d) and further corroborated in non-cancerous RPE1 cells (Supplementary Fig. 3e-h). Moreover, the ablation of the *DCAF12* gene by the CRISPR-Cas9 knockout approach led to the stabilization of endogenous MCMBP protein levels. This effect was reversed by the doxycycline-inducible re-expression of StrepII-Flag-tagged DCAF12 protein in DCAF12-knockout cells, as evidenced by QIBC (Fig. 2a-d, Supplementary Fig. 3i, j). Additionally, a cycloheximide chase experiment demonstrated that MCMBP protein stability was greater in DCAF12-knockout cells, which was again reversible with exogenous DCAF12 re-expression (Fig. 2e-g and Supplementary Fig. 3k, l). Notably, over-expression of StrepII-Flag-tagged DCAF12, greatly exceeding its endogenous levels (Supplementary Fig. 3j), reduced MCMBP protein below its basal levels, indicating precise cellular regulation of DCAF12 to achieve optimal MCMBP levels. Consistent with this, QIBC analysis showed that the increased levels of MCMBP following DCAF12 depletion were predominantly localized in the nucleus and not in the cytoplasm (Fig. 2a-d and Supplementary Fig. 3a-i), aligning with our previous findings that DCAF12 is predominantly a nuclear protein[20]. Moreover, cell-cycle-specific analysis revealed stabilization of MCMBP protein levels in late S/G2 phases upon DCAF12 depletion, indicating specific degradation of MCMBP in these cell cycle phases

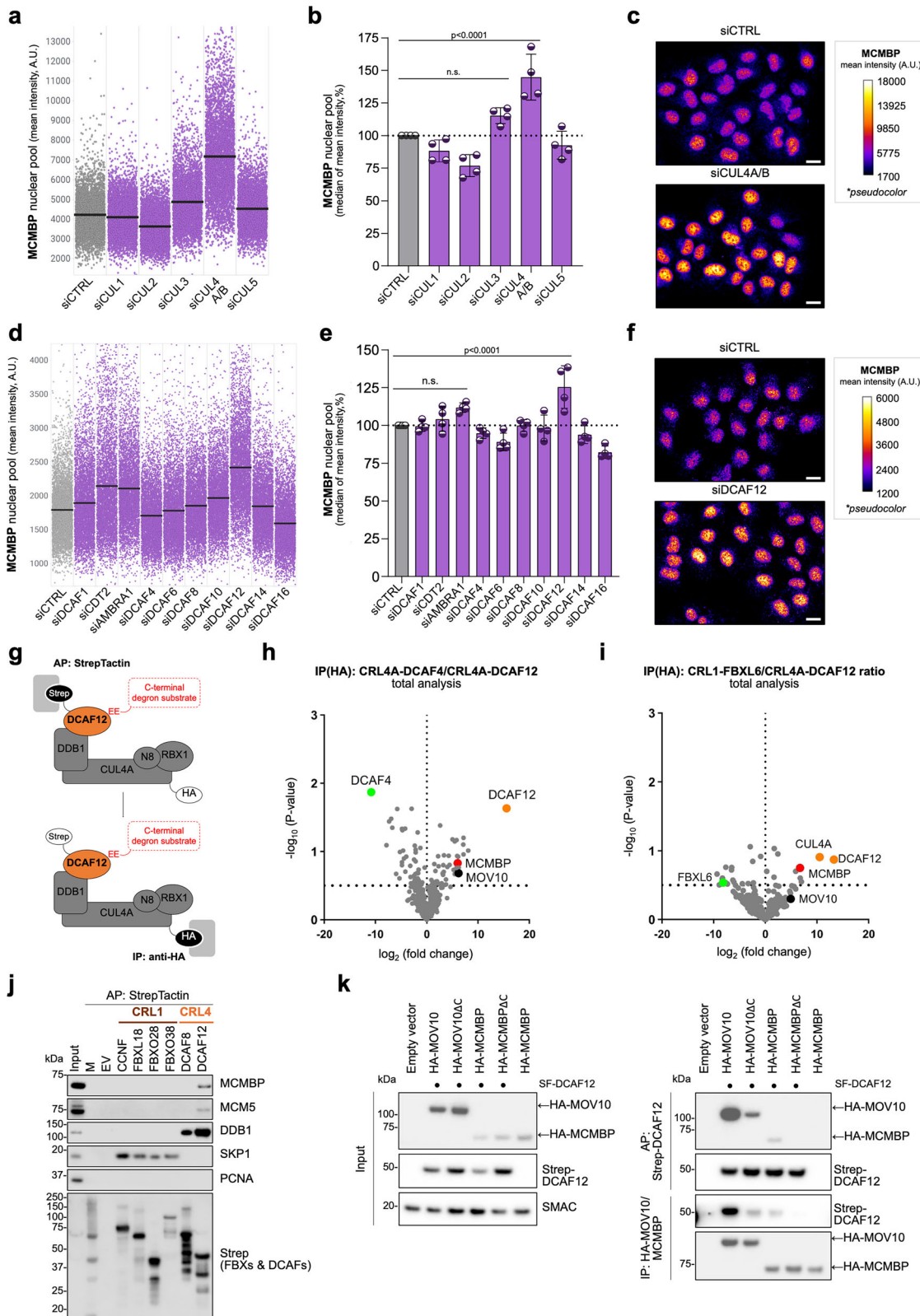

(Fig. 2h, i). This is aligned with MCM biogenesis occurring during the S phase and reaching its maximum in the G2 phase[2,23]. To further demonstrate that DCAF12 facilitates ubiquitin-mediated degradation of MCMBP, we carried out an in vitro ubiquitination assay using affinity-purified DCAF12-containing complexes. In the presence of ATP and recombinant CUL4[NEDD8]–RBX1, MCMBP became poly-ubiquitylated, indicating that DCAF12 actively promotes its

ubiquitination (Fig. 2j). Finally, the QIBC analysis revealed that DCAF12-depleted cells in the presence of proteasome inhibitor MG132 showed a similar stabilization effect on nuclear MCMBP protein levels as in DCAF12-depleted cells without MG132 treatment (Fig. 2k, l). Altogether, we conclude that cellular levels of MCMBP protein are spatiotemporally regulated by CRL4[DCAF12]-mediated ubiquitination and proteasomal degradation.

**Fig. 1 | CRL4$^{DCAF12}$ regulates MCMBP cellular level through direct interaction with its C-terminal degron motif. a, d** QIBC plots of MCM4-Halo cells treated with siRNAs as indicated. Lines denote medians; $n = 4973$ (minimum cells) (**a**) or 5003 (minimum cells) (**d**) per condition. **b, e** Quantification of QIBC plots; each bar indicates the median of mean intensity (data are mean ± s.d.; $n = 4$ (2 biological replicates, each with 2 technical replicates) for both (**b, e**). **c, f** Representative images; the pseudo-color gradient indicates the mean fluorescent intensity (MFI). Scale bar, 20 μm. **g** Schematic representation of the tandem affinity purification strategy. **h, i** CRL4$^{DCAF12}$-associated proteins identified by liquid chromatography-tandem mass spectrometry (LC-MS/MS). Non-related ubiquitin ligases CRL4$^{DCAF4}$ in (**h**) or SCF$^{FBXL6}$ in (**i**) were used as a control. Proteins that were significantly enriched with CRL4$^{DCAF12}$ compared to controls are in the upper right quadrants ($n = 2$ biological replicates). **j** Affinity purification (AP) of a panel of CRLs. HEK293T cells were transfected with an empty vector (EV) or StrepII-FLAG-tagged CRL constructs for 48 h and treated with MLN4924 for 6 h before harvesting. Whole-cell lysates were subjected to AP with Strep-TactinXT resin. M denotes a molecular marker. **k** Affinity purification of N-terminal StrepII-FLAG-tagged DCAF12 (SF-DCAF12) and immunopurification of HA-tagged MCMBP, MOV10, and C-terminal degron-lacking mutants. HEK293T cells were co-transfected with SF-DCAF12 and either an empty vector or specified gene constructs. MLN4924 was added for the final 6 h before harvest. Whole-cell lysates were subjected to affinity purification (AP) with Strep-TactinXT resin or anti-HA immunoprecipitation (IP). The left panel shows 1% input samples, the upper right panel shows Strep-Tactin AP, and the lower right panel shows anti-HA IP. **j, k** are representatives of three independent replicates with similar results. For (**h, i**), statistical analysis was performed in Perseus (version 1.6.15.0) using a two-sample Student's t test (two-sided; $S_0 = 0.1$) with permutation-based false discovery rate (FDR) correction (FDR = 0.05). P values were calculated by ordinary one-way ANOVA with Dunnett's test (**b, e**); n.s. (not significant) indicates $p > 0.1$. (A.U. Arbitrary Units). Source data are provided as a Source data file.

## Epistatic relationship between CRL4$^{DCAF12}$ and MCMBP in the MCM biogenesis pathway

Recent studies have reported that MCMBP acts as the specific chaperone of nascent MCM helicases, and its cellular levels and activity are needed for optimal origin licensing[2,11]. Indeed, reduced levels of MCMBP were shown to impair MCM loading on chromatin[2,11]. Therefore, it is reasonable to anticipate that elevated MCMBP levels resulting from DCAF12 depletion would lead to increased origin licensing. However, the depletion of DCAF12 unexpectedly led to reduced chromatin binding of all MCM subunits (Supplementary Fig. 4a–l). This phenotype was further recapitulated with two independent siRNAs against the *DCAF12* gene (Supplementary Fig. 4m–o) and was not caused by the cell cycle changes (Supplementary Fig. 5a, b). To confirm the potential role of DCAF12 in the stability and loading of nascent MCMs on chromatin, a dual HaloTag labeling protocol (Supplementary Fig. 5c) was employed to distinguish the chromatin binding of parental and nascent MCM complexes. While the loading of parental MCMs on chromatin was not altered by depletion of DCAF12 or MCMBP, the chromatin licensing of nascent MCMs was impaired in both DCAF12- and MCMBP-depleted cells (Fig. 3a–d). Notably, the simultaneous depletion of DCAF12 alongside MCMBP did not exacerbate the inability to load the optimal levels of nascent MCM complexes (Fig. 3e–g). These findings suggest that CUL4-DCAF12 is not responsible for the degradation of the freely available MCMBP protein. Instead, they imply a complex regulatory mechanism wherein DCAF12 modulates MCMBP chaperone activity during the biogenesis of nascent MCM complexes occurring from the S phase until the G2 phase[2,23]. This is also supported by the S/G2-phases-specific regulation of MCMBP levels by DCAF12 (Fig. 2h, i). Collectively, these observations reinforce the notion that MCMBP is a specific chaperone for nascent MCM complexes and identify CRL4$^{DCAF12}$ as a regulator of the MCM biogenesis pathway.

## DCAF12 depletion impairs origin licensing but not localization or total nuclear levels of MCMs

Before delving into the role of CRL4$^{DCAF12}$ in MCM2-7 production, we shortly summarize the key findings that lead us to propose a model for the MCM biogenesis pathway (Fig. 4a). The presented model is built on the previous experiments revealing that shortly after MCMBP depletion or mutating its nuclear localization signal (NLS), nascent MCM3-7 subunits are rapidly mislocalized in the cytoplasm, where they undergo proteasomal degradation[2]. Compared to MCM3-7 subunits, MCM2 remains orphaned in the nucleus, and its levels do not change. These observations align with previous studies showing a strong association of MCMBP with MCM3-7 subunits, as opposed to a weak interaction with MCM2[24,25]. Based on this, we postulate a model in which MCMBP facilitates the assembly of the nascent MCM3-7 subcomplex and contributes to its transport to the nucleus, while the MCM2 subunit autonomously enters the nucleus. Once all subunits are

in the nucleus, the entire MCM2-7 ring is assembled and can be loaded onto chromatin by licensing factors during the G1 phase (Fig. 4a).

To identify the specific step of the MCM biogenesis pathway regulated by CRL4$^{DCAF12}$, we examined the assembly of nascent MCM subcomplexes in the cytoplasm and their transport to the nucleus. Using a dual HaloTag labeling protocol (Supplementary Fig. 5c) and QIBC, we tracked the levels of parental and nascent MCM complexes across both cellular compartments. Consistent with our previous analysis of parental MCM loading on chromatin (Fig. 3a, b), we observed that neither nuclear nor cytoplasmic levels of parental MCMs were affected by DCAF12 or MCMBP depletion (Supplementary Fig. 5d–g). Notably, QIBC analysis revealed that MCMBP depletion resulted in reduced nuclear levels of nascent MCMs due to their mislocalization in the cytoplasm (Fig. 4b–f), where they ultimately undergo proteasomal degradation[2]. In contrast, DCAF12 down-regulation did not impact the overall levels of nascent MCMs in either the nucleus or the cytoplasm (Fig. 4b–f). In fact, no change in nuclear and cytoplasmic levels upon DCAF12 depletion was observed for all MCM subunits (Supplementary Fig. 5h, i). These findings indicate that CRL4$^{DCAF12}$ does not regulate the assembly of the MCM3-7 subcomplex or its transport to the nucleus (steps i and ii in Fig. 4a) but is likely involved in downstream steps of this pathway. The experiments in the following paragraphs indicate that this is indeed the case.

To further reinforce our conclusions and eliminate the possibility that the observed phenotype of MCM loading following DCAF12 depletion is an indirect consequence of prolonged exposure to siRNA treatment, we seek to develop a cell system that enables the examination of the effect of DCAF12 on MCM levels and their loading on chromatin within shorter time periods. Despite several unsuccessful attempts to produce the functional auxin-inducible degron of DCAF12, we leveraged MCMBP-KO cells generated in our earlier study[2]. In this genetic background, we prepared two stable cell lines re-expressing HA-tagged MCMBP wild-type and a mutant lacking the -EL* C-terminal degron motif (designated as ΔC), essential for interaction with DCAF12, under the control of a doxycycline-inducible promoter. Intriguingly, 24-h doxycycline treatment of these cells showed that both wild-type MCMBP and MCMBP-ΔC mutant equally restored nuclear and cytoplasmic levels of MCM4-Halo (Fig. 4g–i). Nevertheless, only wild-type MCMBP, not the MCMBP-ΔC, rescued the chromatin loading of MCM4-Halo (Fig. 4j–l). These observations with inducible systems align with our siRNA findings, suggesting that CRL4$^{DCAF12}$ is not involved in the assembly of the MCM3-7 subcomplex or its nuclear transport, but it may regulate the MCMBP binding during the assembly of the nascent MCM2-7 complex (step iii in Fig. 4a).

## CRL4$^{DCAF12}$ regulates the removal of MCMBP during the assembly of nascent MCM2-7 complexes

As shown previously, MCMBP functions as a dedicated chaperone for the nascent MCM helicases, but it is not part of pre-replicative

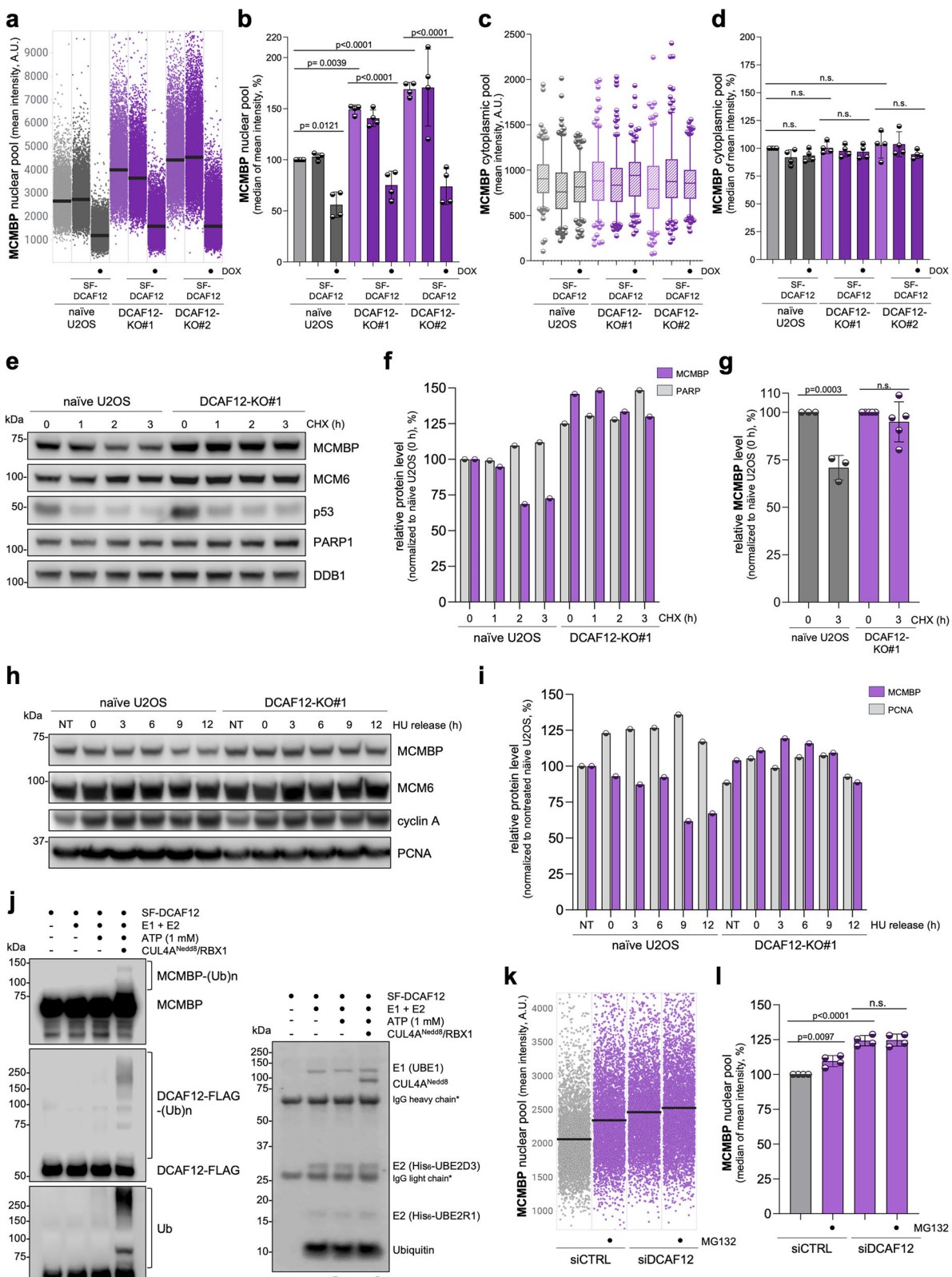

complexes loaded on chromatin[2,11]. This underscores the necessity to tightly regulate the interaction of MCMBP with MCMs. If the removal of MCMBP is either impaired or delayed, it may hinder the assembly of the complete MCM2-7 ring. Consequently, this would lead to a reduced MCM2-7 loading on chromatin, while the localization and total nuclear levels of MCM2-7 would remain unaffected, as indicated by the experiments in Figs. 3 and 4. Based on this, we hypothesized

that CRL4[DCAF12] may regulate the interaction of MCMBP with MCMs during the assembly of a complete MCM2-7 ring (step iii in Fig. 4a). To test this, we monitored the interaction between MCMBP and MCM subcomplexes in control and DCAF12-depleted cells using immuno-precipitation. We considered two potential scenarios: (1) MCMBP and MCM2 are mutually exclusive, implying that MCMBP needs to be removed in order to incorporate MCM2 into the MCM3-7 subcomplex

**Fig. 2 | MCMBP protein levels are spatiotemporally regulated by CRL4[DCAF12]-mediated ubiquitination and proteasomal degradation. a** QIBC plots of U2OS and DCAF12 knock-out cells (DCAF12-KO#1, DCAF12-KO#2) expressing doxycycline (DOX, 48 h) inducible StrepII-FLAG-tagged DCAF12 (SF-DCAF12). Lines denote medians; *n* = 9198 (minimum cells per condition). **b** Quantification of (**a**); each bar indicates the median of mean intensity (data are mean ± s.d.; *n* = 4 (2 biological with 2 technical replicates). **c** MFI of similar conditions as in (**a**). Central lines of box plots denote medians, the boxes indicate the 25th and 75th centiles, the whiskers indicate 5 and 95 percentile values; *n* = 200 cells per condition. **d** Quantification of (**c**); each bar indicates the median of mean intensity normalized to naïve U2OS as 100% (data are mean ± s.d.; *n* = 4 (2 biological with 2 technical replicates). **e** U2OS and DCAF12 knock-out cells were treated with cycloheximide (CHX) as indicated. Soluble protein extracts were immunoblotted. **h** U2OS and DCAF12 knock-out cells were synchronized in G1/S by hydroxyurea (HU, 2 mM; 16 h) and then released in fresh medium for the indicated hours. Whole-cell lysates were immunoblotted. **f, i** Quantification of MCMBP and PARP staining (**f**; from **e**) and MCMBP and PCNA

staining (**i**; from **h**). Each bar indicates relative protein levels normalized to naïve U2OS (0 h) (**f**) or non-treated (NT) naïve U2OS (**h**) as 100%. Data are absolute values from *n* = 1 biological replicate (**f, i**). **g** Quantification of relative MCMBP levels at 0 and 3 h after CHX treatment based on blots in (**e**), Supplementary Fig. 3k, and an additional replicate (see Source data). Each bar indicates protein levels normalized to 0 h timepoint in naïve U2OS (*n* = 3 biological replicates) or DCAF12-KO cells (*n* = 5 biological replicates) as 100%. Data are presented as mean ± s.d. **j** In vitro ubiquitination assay using StrepII-FLAG-tagged DCAF12 (SF-DCAF12) affinity-purified complexes. **k** QIBC plots of MCM4-Halo U2OS cells treated with MG123 inhibitor and siRNAs, as indicated. Lines denote medians; *n* = 4958 (minimum cells per condition). **l** Quantification of (**k**); each bar indicates the median of mean intensity normalized to siCTRL as 100% (data are mean ± s.d.; *n* = 4 (2 biological with 2 technical replicates). *P* values were calculated by ordinary one-way ANOVA with Tukey's test (**b, d**); or Šidák's test (**g**), n.s. indicates *p* > 0.1. (DOX doxycycline, h hours, NT non-treated). Source data are provided as a Source data file.

(2) MCMBP and MCM2 are not mutually exclusive, meaning that they occupy different structural surfaces in the MCM3-7 subcomplex. However, to achieve the complete formation of the MCM2-7 complex that can be loaded on chromatin, MCMBP needs to be removed from the MCM2-7 structure (Fig. 5a). If the first scenario holds true, then DCAF12-depleted cells should manifest enhanced interaction between MCMBP and the MCM3-7 subcomplex and diminished interaction between MCM2 and MCM3-7. If the second scenario is valid, we should observe an increased interaction between MCMBP and MCM3-7 and an unchanged interaction between MCM2 and MCM3-7. The immunoprecipitation of MCM4-Halo did indeed show an increased association of MCMBP with MCMs upon DCAF12 depletion (Fig. 5b, c). This effect was further pronounced when immunoprecipitation was repeated with only the nuclear fractions of MCM4-Halo (Fig. 5d, e). Importantly, the interaction with MCM2 remained unchanged, even when using a higher salt concentration in the immunoprecipitation reaction (Supplementary Fig. 5a, b). To further support our immunoprecipitation findings, we employed proximity ligation assays (PLA) to examine the interaction between MCMBP and MCM2. In agreement with our model, DCAF12 depletion led to increased PLA signal (Fig. 5f, g), indicating that MCMBP remains associated with nascent MCM2-7 complexes when DCAF12 is absent. We next assessed the interaction between CDT1 and MCM2 during G1 phase, using it as a proxy for the formation of mature, chromatin-loading competent MCM2-7 complexes during origin licensing. Notably, the CDT1-MCM2 PLA signal was reduced in DCAF12-depleted cells (Fig. 5h, i), suggesting that loss of DCAF12 impairs proper maturation of the MCM2-7 complex. To further confirm that DCAF12 recognizes MCMBP when it is part of the MCM2-7 complex, we analyzed the DCAF12 interactome under normal conditions and after MG132 treatment (Fig. 5j; Supplementary Fig. 6c). Our analysis revealed that DCAF12 interacts with the MCMBP-MCM2-7 complex, with MCM2 interaction becoming more prominent upon MG132 treatment. These findings indicate that MCMBP and MCM2 lack mutually exclusive binding surfaces within the MCM structure and suggest that CRL4[DCAF12] facilitates the removal of MCMBP from the MCM2-7 complex.

Throughout our study, we demonstrated that stabilization of MCMBP upon DCAF12 depletion is confined to the cell nucleus and not the cytoplasm. This suggests that CRL4[DCAF12] regulates MCMBP in the context of the MCM2-7 complex within the nucleus, which is consistent with previous findings identifying DCAF12 as a nuclear protein[20]. To further support our observations, we utilized our previously published cell systems HCT116 expressing DCAF12 variants, including wild-type and mutants lacking one or two NLS motifs (Fig. 5k; Supplementary Fig. 6d)[20]. Although the mutants lacking NLS were still detected in the nuclear fraction, presumably due to the involvement of other proteins in DCAF12 nuclear transport, we observed an approximately threefold increase in cytoplasmic signal of DCAF12 mutants in

comparison to wild type (Fig. 5l; Supplementary Fig. 6e). To control functionality of this system, we used GART protein that has previously been shown to be exclusively located in the cytoplasm and its C-terminal degron motif is recognized by DCAF12 when it is forced into the cytoplasm (Supplementary Fig. 6d)[20]. In this experimental system, we sought to investigate whether the forced mislocalization of DCAF12 in the cytoplasm would also lead to the degradation of MCMBP in this cellular compartment. QIBC analysis revealed that DCAF12 does not degrade MCMBP in the cytoplasm but targets it for degradation exclusively in the nucleus (Fig. 5m; Supplementary Fig. 6f). This suggests that during the assembly and transport of MCM3-7 subcomplexes, the acidic tail of MCMBP may be shielded from DCAF12. Analogous to the behavior of the acidic tail of CCT5, a component of the chaperonin complex under the regulation of DCAF12, the acidic tail of MCMBP may be concealed within the MCM3-7 structure[26]. Once in the nucleus, conformational changes during the final MCM2-7 complex assembly might release the acidic tail, allowing it to be recognized by DCAF12. Such a regulatory mechanism, facilitated by CRL4[DCAF12], further refines the chaperone function of MCMBP while concurrently introducing an additional quality control layer into the process of assembling nascent MCM complexes.

## DCAF12 is an important regulator of optimal MCM equilibrium

Using various cellular systems and approaches to manipulate DCAF12 protein in this study, we noted that tight cellular regulation of DCAF12 is essential to achieve optimal MCMBP levels and, thus, proper MCM equilibrium. We showed that the downregulation of DCAF12 leads to impaired removal of MCMBP during the MCM2-7 complex assembly (Fig. 5a–i), which phenotypically appears as increased MCMBP levels and reduced origin licensing (Supplementary Fig. 3a, b; Supplementary Fig. 4a–o). Nevertheless, the up-regulation of DCAF12 levels also poses a threat to optimal MCMBP levels. Indeed, overexpression of DCAF12, approximately tenfold above its endogenous levels, led to substantial degradation of MCMBP below its basal levels (Fig. 2a, b; Supplementary Fig. 3j). This, in turn, caused the massive degradation of MCM complexes in an MCMBP-dependent manner (Supplementary Fig. 6g–j), which, as shown previously, results in impaired origin licensing[2]. Interestingly, previous proteomic datasets showed that CUL4A/B are highly abundant proteins in the cellular environment, reaching a similar copy number as MCM proteins[27]. However, the cellular level of DCAF12 protein is about a hundred times lower than the amount of CUL4A/B. Our findings demonstrated that too little or too high cellular levels of DCAF12 are detrimental to origin licensing, explaining the necessity to tightly regulate DCAF12 protein level, reminiscent of the well-known Goldilocks principle, to ensure its fine specificity under physiological conditions. Based on this, we conclude that DCAF12 is a critical regulator of the MCM biogenesis pathway in mammalian cell systems.

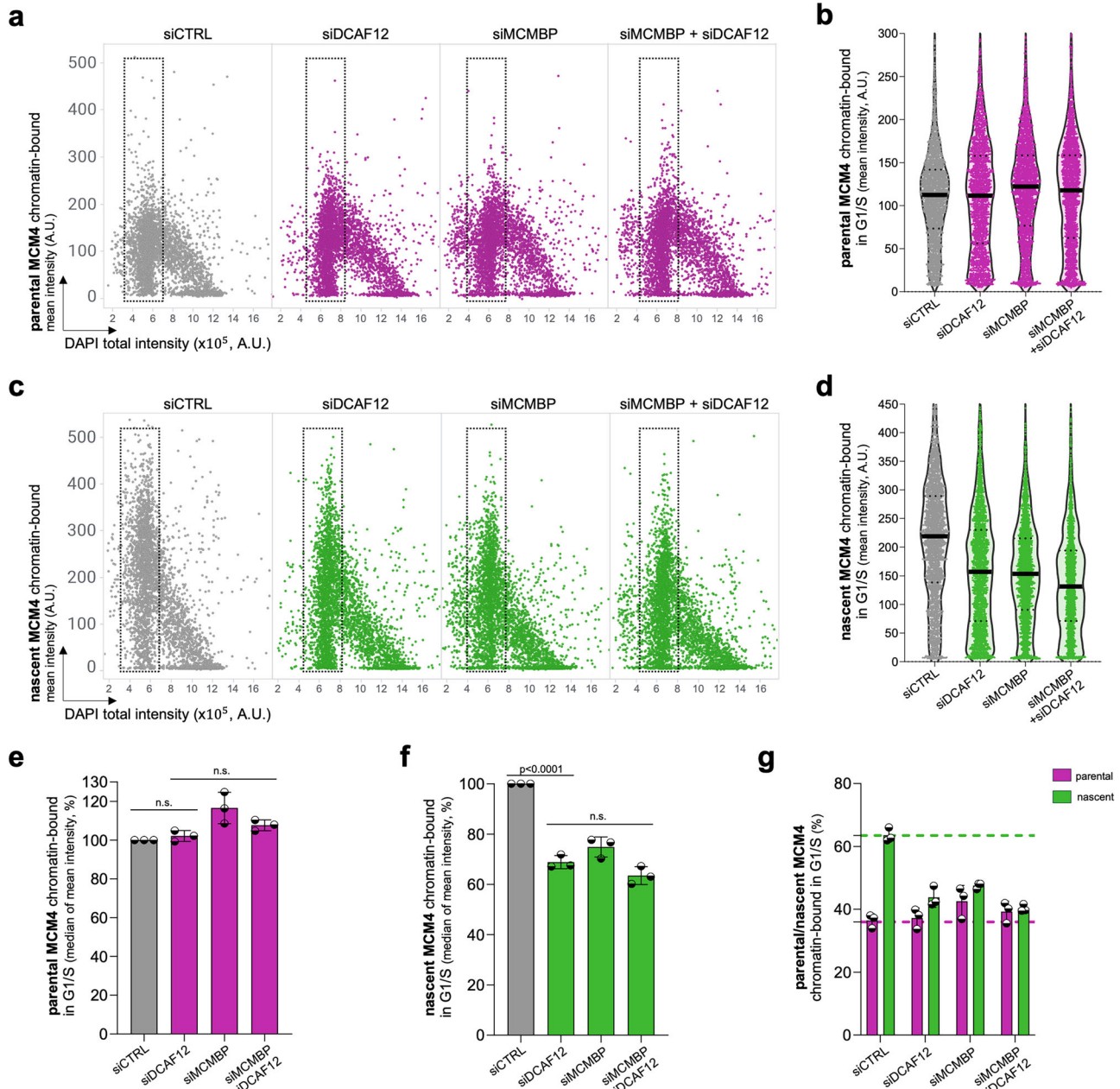

**Fig. 3 | CRL4$^{DCAF12}$ is a key regulator of the MCM biogenesis pathway. a, c** QIBC plots of pre-extracted MCM4-Halo U2OS cells stained for parental and nascent MCM4 after indicated siRNA treatments. DAPI counterstains nuclear DNA; $n = 3932$ (minimum cells per condition). **b, d** MFI of parental and nascent MCM4 in G1/S cells based on QIBC in (**a, c**), respectively. Lines denote medians; $n = 2029$ (minimum cells per condition). **e, f** Quantification of the MFI of parental and nascent MCM4 (in G1/S) in (**b, d**), respectively. Each bar indicates the median of mean intensity

normalized with respect to siCTRL as 100% (data are mean ± s.d.; $n = 3$ biological replicates). **g** Quantification of chromatin-bound levels of parental and nascent MCMs in G1/S phase MCM4-Halo cells after indicated siRNA treatments (data are mean ± s.d.; $n = 3$ biological replicates). $P$ values were calculated by ordinary one-way ANOVA with Šidák's test (**e, f**), n.s. indicates $p > 0.1$. Source data are provided as a Source data file.

## Misregulation of the MCM biogenesis pathway leads to genome instability

Prior research demonstrated that even minor imbalances in MCM equilibrium can lead to genome instability and severe diseases, including cancer[2,4–8]. Although nascent MCMs remain largely in the inactive chromatin-bound state during the DNA replication program, they serve as natural replisome pausing sites, helping to coordinate replisome movement through chromatin[2,28,29]. These pausing sites are part of the multilayer replication fork speed control, which recently emerged as a pivotal genome surveillance mechanism protecting cells against replication stress and genome instability[30–33]. Notably, reducing

natural replisome pausing sites by direct depletion of nascent MCMs or their chaperone MCMBP led to accelerated replication fork speed and generation of DNA damage without a major impact on CMG formation and the origin firing program[2]. In light of these findings, we sought to explore whether impaired origin licensing caused by the down-regulation of DCAF12 leads to similar hallmarks of replication stress. Using the DNA fiber method, we observed that the depletion of DCAF12 resulted in increased replication fork speed without alteration of chromatin-bound CDC45 or PCNA (Fig. 6a; Supplementary Fig. 7a–e). Similarly to MCMBP depletion, unrestrained fast fork progression due to DCAF12 depletion led to an elevated incidence of fork asymmetry,

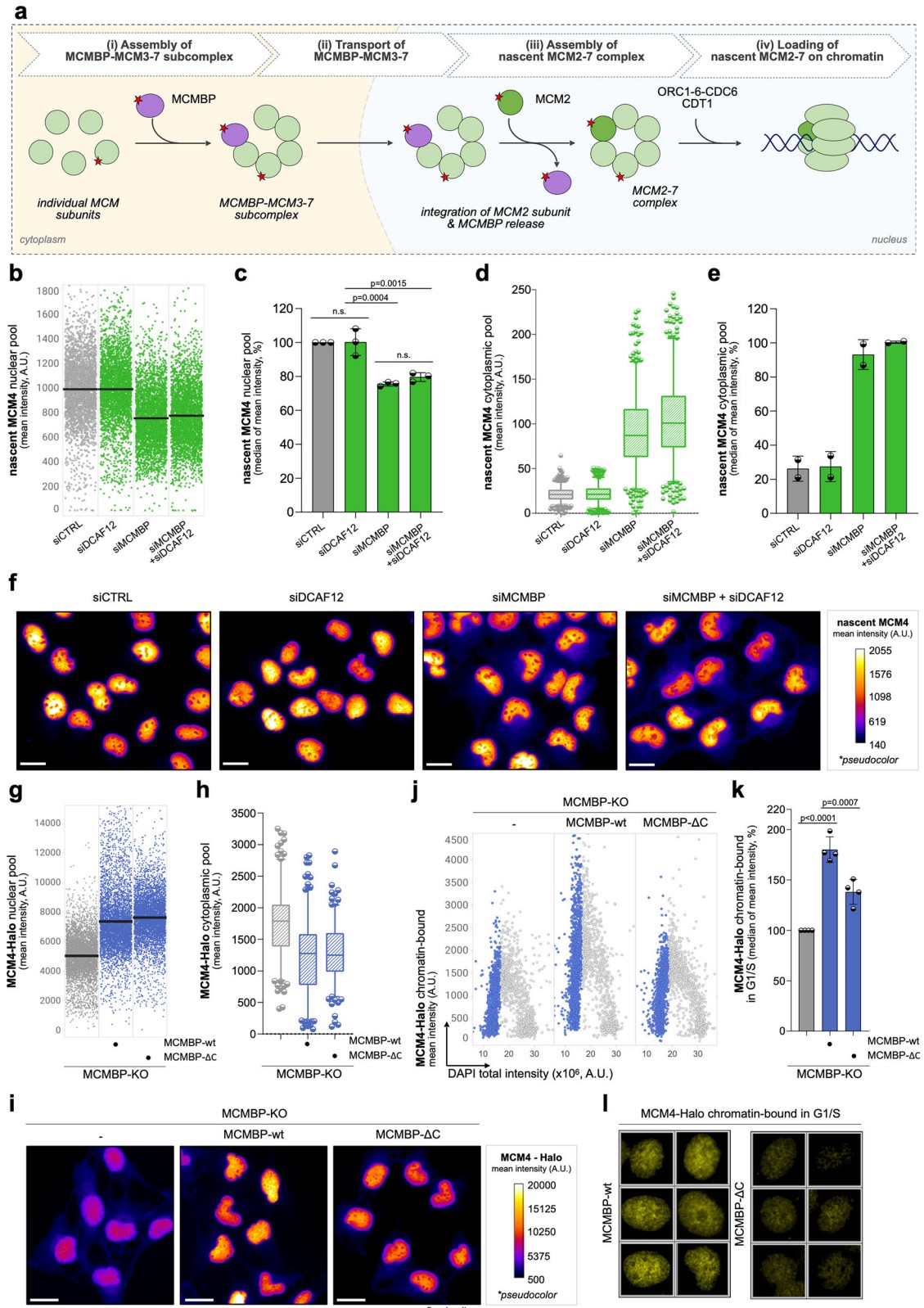

pointing toward a higher frequency of replication fork pausing or stalling events (Fig. 6b). Importantly, neither the replication fork speed nor the fork asymmetry phenotypes were exacerbated when DCAF12 and MCMBP were co-depleted, supporting the notion that DCAF12 and MCMBP operate within the same signaling pathway (Fig. 6a, b).

As shown previously, loss of replication fork speed control can lead to replication-born DNA damage[2,30,31]. Our QIBC analysis revealed

an elevated level of γH2AX during the S phase, alongside an increased number of spontaneous RAD51 foci, well-established markers of DNA damage (Fig. 6c, d; Supplementary Fig. 7f–j). Notably, the generated DNA damage can be mitigated by re-expression of MCMBP wild type, but not MCMBP-ΔC mutant, which lacks the DCAF12 binding site and therefore cannot restore origin licensing in MCMBP-KO cells (Figs. 4j, k; 6e). Previous studies have demonstrated that increased

**Fig. 4 | The assembly of the nascent MCM3-7 subcomplex or its transport to the nucleus is not affected by DCAF12 depletion. a** Proposed model of MCM biogenesis pathway (see text for details). **b** QIBC plots of MCM4-Halo U2OS cells stained for nascent MCM4 after indicated siRNA treatments. Lines denote medians; $n = 3351$ (minimum cells per condition). **c** Quantification of QIBC plots in (**b**); each bar indicates the median of mean intensity (data are mean ± s.d.; $n = 3$ biological replicates). **d** MFI of cytoplasmic nascent MCM4 after indicated siRNA treatments. Central lines in box plots denote medians, the boxes indicate the 25th and 75th percentiles, the whiskers indicate 5 and 95 percentile values; $n = 500$ cells per condition. **e** Quantification of MFI of cytoplasmic nascent MCM4; each bar indicates the median of mean intensity (data are mean ± s.d.; $n = 2$ biological replicates). **f** Representative images from (**b, d**); the pseudo-color gradient indicates the MFI. Scale bar, 20 μm. **g** QIBC of MCMBP knock-out U2OS (MCMBP-KO) cells expressing wild-type MCMBP (MCMBP-wt) or C-terminal deleted MCMBP (MCMBP-ΔC) after

24 h of DOX induction, stained for MCM4-Halo (without pre-extraction). Lines denote medians; $n = 5018$ (minimum cells per condition). **h** MFI of cytoplasmic MCM4-Halo under similar conditions as in (**g**); Central lines in box plots are medians, the boxes indicate the 25th and 75th centiles, the whiskers indicate 5 and 95 percentile values; $n = 210$ (minimum cells per condition). **i** Representative images from (**g, h**); the pseudo-color gradient indicates the MFI. Scale bar, 20 μm. **j** QIBC plots of indicated cells (DOX, 24 h) stained for MCM4-Halo after pre-extraction. DAPI counterstained nuclear DNA; $n = 2957$ (minimum cells per condition). **k** Quantification of QIBC in (**j**); each bar indicates the median of mean intensity normalized with respect to MCMBP-KO as 100%; data are mean ± s.d.; $n = 4$ (2 biological clones of indicated cell lines with 2 technical replicates). **l** Unbiased QIBC galleries from (**j**). $P$ values were calculated by ordinary one-way ANOVA with Tukey's test (**c, k**), n.s. indicates $p > 0.1$. Source data are provided as a Source data file.

replication stress can lead to genome instability[2,30–33]. This phenomenon is also observed in DCAF12-depleted cells, as indicated by abnormalities in cell division and elevated levels of micronuclei formation (Fig. 6f and Supplementary Fig. 7k). Ultimately, the accumulation of genome instability over multiple generations of diving cells may account for the reduced colony formation observed in DCAF12 and MCMBP-depleted cells (Fig. 6g). Collectively, our findings suggest that precise regulation of MCMBP chaperone function through CRL4[DCAF12] is important to sustain optimal levels of inactive nascent MCMs on chromatin, essential to preserve physiological replication fork speed and genome stability (Fig. 6h).

## Discussion

The precise maintenance of cellular MCM equilibrium is crucial to ensure error-free genome duplication. Our current work provides further insights into the molecular mechanism of the MCM biogenesis pathway, essential for optimal loading of nascent MCM on chromatin, thereby mitigating replication stress and genome instability (Fig. 6h). Building on our current and previous findings[2], we propose a model in which MCMBP, a dedicated chaperone for nascent MCM complexes, fosters the assembly of nascent MCM3-7 subcomplexes in the cytoplasm and facilitates their translocation to the cell nucleus. In contrast to the MCM3-7 subcomplex, the cellular level and nuclear transport of the MCM2 subunit are independent of MCMBP. Although the reason for the separation of the MCM complex during nuclear transport is not yet clear, a recent study suggested that certain protein cargos can enhance the mechano-selectivity of the nuclear pore complex and facilitate nuclear import rates[34]. Further elucidation will be needed to determine if MCMBP may potentially possess analogous attributes.

After the MCMBP delivers MCM3-7 subcomplexes into the nucleus, the MCM2 subunit needs to be integrated into the MCM complex structure, while the MCMBP must be removed as it is not part of pre-replicative complexes loaded on chromatin[2,11]. This highlights the need to tightly regulate the interaction between the MCMBP and MCMs. In this work, we have uncovered that CRL4[DCAF12] is a bona fide regulator of this interaction (Fig. 6h). Our findings suggest that CRL4[DCAF12] recognizes and interacts with the C-terminal degron motif of MCMBP, resulting in the ubiquitination and proteasomal degradation of MCMBP. Specifically, we have demonstrated that CRL4[DCAF12] facilitates the removal of MCMBP from the MCM2-7 complex in the nucleus, representing an important step in the assembly of a complete MCM2-7 ring that can be subsequently loaded onto chromatin during origin licensing. Notable is our observation that the activity of DCAF12 does not impede the assembly and transportation of MCM3-7 subcomplexes, even when DCAF12 is artificially delocalized to the cytoplasm (Fig. 5k–m; Supplementary Fig. 6d–f). This suggests that during these steps, the acidic tail of MCMBP may be shielded from DCAF12, similar to the protective mechanism described previously for the acidic tail of CCT5, a component of the chaperonin complex regulated

by DCAF12[26]. Upon delivery of the MCMBP-MCM3-7 complex to the nucleus, it is feasible that engagement of the MCM2 subunit with the complex can induce a conformational alteration in the MCMBP binding. This, in turn, may facilitate the release of the acidic tail, enabling its recognition by DCAF12. However, further structural studies are necessary to gain detailed insights into the assembly of the MCM2-7 complex, particularly in terms of the molecular interactions between MCMBP and this complex. Current experimental studies diverge in this regard, with one study reporting a direct interaction between MCMBP and MCM3 and another reporting a direct interaction between MCMBP and MCM7[11,25]. In addition to structural insights, exploring the evolutionary perspective on MCMBP function could be another exciting area for future research. MCMBP chaperone function during assembly of MCM complexes appears to be conserved from fission yeast to higher eukaryotes[35], prompting the need to investigate why MCMBP is absent in budding yeast. Notably, certain species, such as *Schizosaccharomyces pombe* or *Arabidopsis thaliana*, possess MCMBP and CUL4 but lack DCAF12 (Supplementary Fig. 2h). Our experiments measuring the stability of MCMBP in DCAF12 knockout cells indicate a certain degree of adaptability in this process (Supplementary Fig. 3k, l), suggesting that, in the absence of DCAF12, alternative biochemical pathways can govern this process. Exploring how the interaction between MCMBP and MCM complexes is regulated in different species represents an intriguing avenue for future research.

Previous research has underscored the crucial roles of various cullin E3 ubiquitin ligases in regulating genome duplication. For instance, CRL4[CDT2] plays a key role in degrading CDT1 after MCM loading on chromatin is completed, preventing re-replication and genome instability[10,17,18]. Furthermore, CUL2[LRR1] has been identified as a driver of replisome disassembly during replication termination[36,37]. Our work contributes to this exciting area by understanding the role of CRL4[DCAF12] in regulating the MCMBP chaperone function during the assembly of nascent MCM2-7 complexes. Additionally, this tight regulation of MCMBP interaction introduces an additional quality control layer in the maintenance of cellular MCM equilibrium important to maintain genome stability. We envisage that future studies exploring the regulatory pathways governing replication protein homeostasis may yield valuable insights into error-free genome duplication and bring new cancer treatment strategies. Given the frequent overproduction of MCMs in various cancer types[38], along with supporting evidence for DCAF12 tumor suppressor activity[39,40], the strategic targeting of MCM equilibrium holds promise for cancer therapy. We hypothesize that pharmacological inhibition of the MCM biogenesis pathway may impair origin licensing, leading to increased replication stress attributable to accelerated fork progression. While normal cells can activate the licensing checkpoint and temporarily arrest in the G1 phase, cancer cells replicate their DNA with reduced replication origins and accumulate DNA damage[3]. Consequently, the amplified burden of DNA damage in cancer cells may render them more susceptible to currently available treatments.

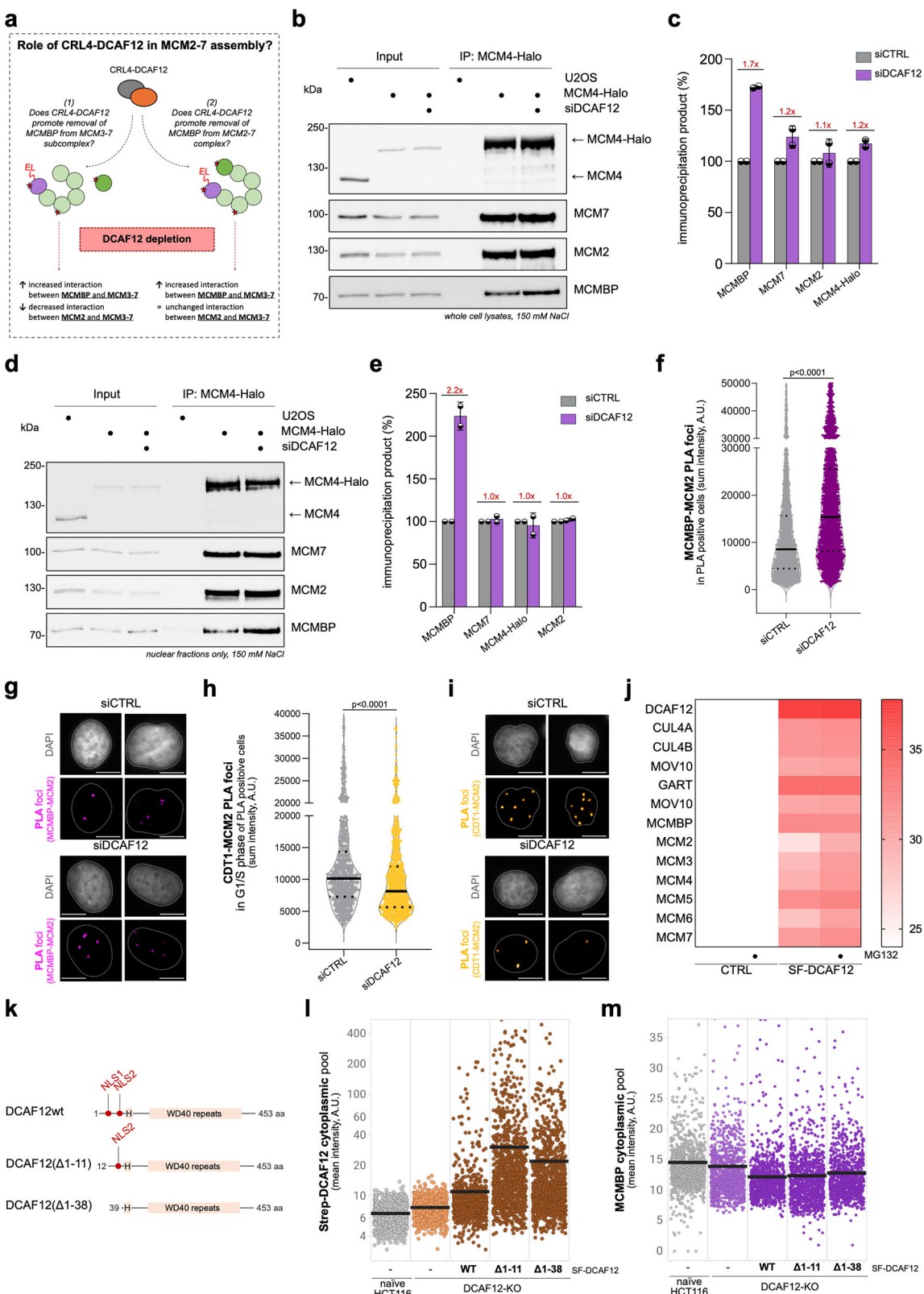

## Methods

### Cell culture

The human immortalized retinal epithelial cell line hTERT-RPE1 (ATCC, CRL-4000), osteosarcoma cancer cell line U2OS (ATCC, HTB-96), and its derivatives were cultured under standard sterile conditions at 37 °C and 5% $CO_2$ level in Dulbecco's modified Eagle's medium (DMEM) containing high glucose and GlutaMAX (Thermo Fischer Scientific,

31966047) supplemented with 10% Fetal Bovine Serum (FBS; Thermo Fischer Scientific, A5256801) and 0.5% penicillin-streptomycin (Thermo Fischer Scientific, 15140122). The human kidney cancer cell line HEK293T (ATCC, CRL-1573), colon cancer cell line HCT116 (ATTC, CCL-247), and its derivatives were maintained in Dulbecco's Modified Eagle's Medium (DMEM) supplemented with 10% FBS, and penicillin (BB Pharma), streptomycin (Merck), and gentamycin (Sandoz), and

**Fig. 5 | CRL4[DCAF12] regulates the interaction between MCMBP and nascent MCM complexes. a** A schematic describing the experimental approach for understanding the role of CUL4-DCAF12 in MCM2-7 assembly (see text for details). **b** Halo-immunoprecipitation (Halo-IP) of whole cell lysates. Cells were treated with siRNA against DCAF12 (siDCAF12) as indicated. **c** Quantification of Halo-IP in (**b**); IP products in control cells have been considered as 100% (data are mean ± s.d.; $n = 2$ biological replicates). **d** Halo-IP of nuclear fraction. Cells were treated with siDCAF12 as indicated. IPs in (**b**, **d**) were done in low salt condition (150 mM NaCl). **e** Quantification of Halo-IP in (**d**); IP products in control cells have been considered as 100% (data mean ± s.d.; $n = 2$ biological replicates). **f** Sum fluorescence intensity (SFI) of PLA foci for the MCMBP-MCM2 antibody pair in U2OS cells treated with control or siRNA against *DCAF12*. Lines denote medians; $n = 3454$ (minimum cells per condition). **g** Representative images from (**f**). This experiment was independently repeated two times with similar results. Scale bar, 10 μm. **h** SFI of PLA foci for

the CDT1-MCM2 antibody pair in G1/S phase of PLA-positive U2OS cells treated with control or siRNA against DCAF12. Lines denote medians; $n = 746$ cells per condition. **i** Representative images from (**h**). This experiment was independently repeated two times with similar results. Scale bar, 10 μm. **j** Heatmap, derived from mass spectrometry analysis of HEK293T cells co-transfected with StrepII-FLAG-tagged DCAF12, shows log2 intensities of indicated proteins. Cells were treated with either DMSO (CTRL) or 10 μM MG132 for 6 h prior to protein purification. **k** Schematic representation of human DCAF12 and its N-terminally truncated mutants. Two predicted nuclear localization signals (NLS) and H-box are highlighted. **l, m** QIBC plots of the cytoplasm of HCT116 cells stained for StrepII-FLAG-tagged DCAF12 (SF-DCAF12) (**l**) or MCMBP (**m**) protein after indicated DCAF12 constructs were inducibly expressed. Lines denote medians; $n = 1051$ (minimum cells) (**l**) or 1105 (minimum cells) (**m**) per condition. *P* values in (**f**, **h**) were calculated by two-tailed unpaired *t*-test. Source data are provided as a Source data file.

cultured in a humidified incubator at 37 °C with 5% $CO_2$. All cell lines and their derivatives were routinely tested for mycoplasma using the Mycoplasma Detection Kit (InvivoGen, rep-mys-50) and were always found to be negative.

## Generation of endogenously tagged cell lines and knockouts using CRISPR-Cas9

A derivative of the U2OS cell line expressing C-terminally endogenously Halo-tagged MCM4 and GFP-tagged CDC45 was generated using CRISPR-Cas9 and validated in the previous study[2]. A U2OS cell line containing a knockout of the MCMBP gene (U2OS MCMBP-KO cells) and C-terminal endogenous Halo-tagged MCM4 was prepared and validated in the previous study[2]. A U2OS cell line with a knockout of the *DCAF12* gene was generated using the CRISPR-Cas9 genome editing system. The sgRNAs were designed to target introns flanking exon 5 of the *DCAF12* gene. A mixture of lenti-CRISPR plasmids (pXPR001) encoding these sgRNAs and Cas9 was transfected into the cells. After a 48-h selection with puromycin (1 μg/ml), single-cell clones were selected, and the exon deletions were confirmed by PCR.

## Generation of cell lines with ectopic protein expression

The HEK293T cell line was transiently transfected with pCDNA3-based vectors expressing Strep II-FLAG-tagged substrate receptors, HA-tagged MOV10, MCMBP, or their C-terminal deletion mutants. The stable cell lines expressing Strep II-FLAG-tagged DCAF12 under the control of a doxycycline (DOX)-inducible promoter were generated using the Sleeping Beauty system as described previously[20]. Briefly, cells were transfected with a pSBtet vector containing DCAF12 cDNA or its mutants lacking the NLS, along with the transposase-expressing pSB100X, using Lipofectamine 2000 (Invitrogen) according to the manufacturer's protocol. Positive clones were selected with puromycin 48 h after transfection. Derivatives of HCT116 ectopically expressing StrepTactin-tagged DCAF12, DCAF12(Δ1-11), and DCAF12(Δ1-38) were prepared and validated in the previous study[20].

To generate stable cell lines expressing doxycycline-inducible variants of MCMBP, U2OS MCMBP-KO cells containing endogenously Halo-tagged MCM4 were transfected with either HA-tagged MCMBP-wt or MCMBP-ΔC constructs using Lipofectamine LTX with Plus reagent (Thermo Fisher Scientific, 15338100). After 24 h of post-transfection, the cells were selected with DMEM containing puromycin (0.1 mg per ml, InvivoGen, ant-pr-1) for 3 days and then serially diluted onto 100 mm dishes. The cells were grown under puromycin selection for 12 days. The isolated colonies were individually expanded and treated with doxycycline (1 μg per ml, Merck, D3072) for 24 h to induce expression of MCMBP-wt or MCMBP-ΔC prior to QIBC-based analysis. Afterward, the cells were pulsed with fluorescent HaloTag ligand as described in the section *HaloTag labeling protocol*, and the nuclear pool of labeled MCM4-Halo was analyzed by QIBC.

## Chemical reagents, siRNAs, and antibodies

Chemical reagents were used in this study as follows. Cycloheximide (Merck, C7698, 100 μg/ml, 355 μM), MLN4924 (Medchemexpress, HY-70062, 1 μM), hydroxyurea (Merck, H8627-1G, 2 mM, 16 h), and doxycycline hyclate (Merck, D9891, 0.1 μg/ml, 48 h) were used as indicated in figures and respective figure legends. For experiments with MG132, U2OS cells were treated with MG132 (Merck, 474790-1MG) at a final concentration of 5 μM for 6 h. For cell cycle synchronization, U2OS cells were incubated with 2 mM hydroxyurea (Sigma, H8627-1G) for 16 h.

All siRNA transfections were performed using the Lipofectamine RNAiMAX Transfection Reagent (Thermo Fisher Scientific, 13778075) according to the manufacturer's recommendation. Cells were treated with siRNAs against CUL1, CUL2, CUL3, CUL4A, CUL4B, and CUL5 at a final concentration of 20 nM for 72 h (Dharmacon/Horizon Discovery, L-004086-00-0005, L-007277-00-0005, L-010224-00-0005, L-012610-00-0005, L-017965-00-0005, L-019553-00-0005) according to a previous study[13]. For remaining experiments, siRNAs were procured from Ambion Silencer Select: DCAF1 (s18762), CDT2 (s226738), AMBRA1 (s31112), DCAF4 (s25085), DCAF6 (s31604), DCAF8 (s27089), DCAF10 (s35726), DCAF12#1 (s24628), DCAF12#2 (s24627), DCAF14 (s30012), DCAF16 (s29650), and MCMBP (s36586). All transfections were performed at a final concentration of 10 nM siRNA for 72 h. Ambion negative control #1 (a non-targeting siRNA) was used as the control siRNA.

Primary antibodies used for immunofluorescence (IF) were as follows: CDT1 (rabbit, Abcam, ab202067, 1:2,000), Cyclin D1 (rabbit, Proteintech, 26939-1-AP, 1:1,000), GFP (rabbit, Proteintech, PABG1, 1:5,000), MCMBP (rabbit, Novus Biologicals, NBP1-90746, 1:1,000), MCM2 (rabbit, Proteintech, 10513-1-AP, 1:1,000), MCM3 (mouse, Santa Cruz, sc-390480, 1:1,000), MCM4 (rabbit, Proteintech, 13043-1-AP, 1:1,000), MCM5 (rabbit, Proteintech, 11703-1-AP, 1:1,000), MCM6 (mouse, Novus Biologicals, H00004175-M04, 1:1,000), MCM7 (mouse, Santa Cruz, sc-9966, 1:1,000), PCNA (human, Immuno Concepts, 2037, 1:1,000), RAD51 (rabbit, BioAcademia, 70-012, 1:1,000), Strep II Tag (mouse, Novus Biologicals, NBP2-43735, 1:1,000), γH2AX (Ser139) (rabbit, Abcam, ab81299, 1:1,000).

Primary antibodies used for western blotting were as follows: α-tubulin (mouse, Proteintech, 66031-1-Ig, 1:1,000), β-actin (mouse, Santa Cruz, sc-69879, 1:1,000), CCNA (home-made serum), DDB1 (rabbit, Zymed, 34-2300, 1:1,000), GART (mouse, Santa Cruz, sc-166379, 1:1,000), γH2AX (rabbit, Proteintech, 83307-2-RR, 1:1,000; mouse, Milipore, 05-636, 1:1,000), HA (rabbit, Cell Signaling, 3724, 1:1,000), MCMBP (rabbit, Proteintech, 19573-1-AP, 1:1,000; rabbit, Atlas Antibodies, HPA038481, 1:1,000), MCM2 (rabbit, Proteintech, 10513-1-AP, 1:1,000), MCM2 (rabbit, ABClonal, A1056, 1:1,000), MCM3 (mouse, Santa Cruz, sc-390480, 1:1,000), MCM3 (rabbit, ABClonal, A1060, 1:1,000), MCM4 (rabbit, Proteintech, 13043-1-AP, 1:1,000), MCM5 (rabbit, Proteintech, 11703-1-AP, 1:1,000), MCM6 (rabbit, Proteintech, 13347-2-AP, 1:1,000), MCM7 (mouse, Santa Cruz, sc-9966, rabbit, Proteintech, 11225-1-AP, 1:1,000), p53 (mouse, Santa Cruz, sc-126, 1:1,000),

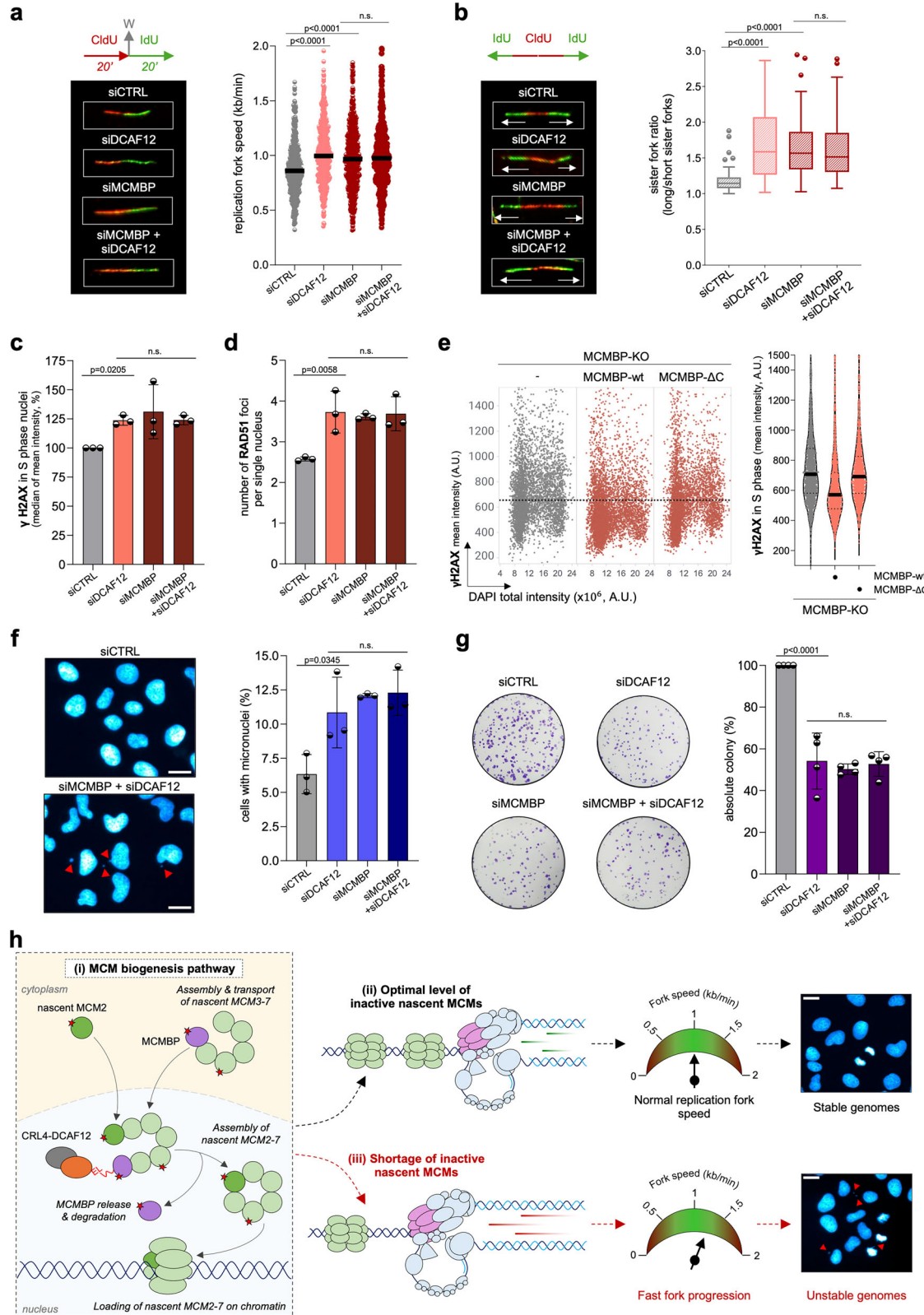

PARP1 (mouse, Proteintech, 66520-1-Ig, 1:1,000), PCNA (mouse, Santa Cruz, sc-56, 1:1,000), pH3 (Ser10) (rabbit, Cell Signaling, 53348, 1:1,000), RPS3A (rabbit, ABClonal, A5885, 1:1,000), RPS6 (mouse, Santa Cruz, sc-74459, 1:1,000), SKP1 (rabbit, Cell Signaling, 12248, 1:1,000).

Secondary antibody conjugates for IF were goat anti-rabbit Alexa Fluor 488 (A11034), goat anti-rabbit and goat anti-mouse Alexa Fluor 568 (A11036, A11031), goat anti-rabbit and goat anti-mouse Alexa Fluor

647 (A21245, A21236) (all from Thermo Fischer Scientific, 1:2,000); donkey anti-human Alexa Fluor 647 (Jackson Immuno Research, 709-605-149, 1:2,000); goat anti-mouse (DyLight 488, Thermo Fisher, 35503, 1:1,000); and donkey anti-rabbit IgG (Alexa Fluor® 555, Abcam, ab150070, 1:1,000). Secondary antibody conjugates used for western blotting were HRP-linked horse anti-mouse IgG (Vector Laboratories, PI-2000, 1:10,000), goat anti-rabbit IgG (Vector Laboratories, PI-1000,

**Fig. 6 | Shortage of nascent MCM complexes on chromatin leads to fast fork progression and genome instability. a** Left, labeling protocol for DNA fiber experiments and a panel of representative images of ongoing replication forks. Right, replication fork speed in U2OS cells treated with the indicated siRNAs. Lines represent medians; $n = 500$ fibers per condition. **b** Left, representative images of bidirectional replication forks; right, quantification of sister fork ratio in U2OS following indicated siRNA treatments. In box plots, central lines denote medians, the boxes indicate the 25th and 75th percentiles, and the whiskers indicate Tukey values; $n = 50$ fibers per condition. **c** Quantification of γH2AX in U2OS after indicated siRNA treatments. Each bar indicates the median of mean intensity normalized to siCTRL as 100% (data are mean ± s.d.; $n = 3$ biological replicates). See QIBC in Supplementary Fig. 7f. **d** Quantification of RAD51 foci in U2OS after indicated treatments. Each bar indicates the median of the mean foci count (data are mean ± s.d.; $n = 3$ biological replicates). See QIBC in Supplementary Fig. 7j. **e** Left, QIBC plots of γH2AX in the indicated U2OS cell lines. DAPI counterstains nuclear DNA; $n = 4955$ (minimum cells per condition). Right, MFI of QIBC on the left. Lines denote median, $n = 1409$ (minimum cells per condition). **f** Representative images (left), quantification (right) of micronucleation in U2OS after indicated treatments (500 nuclei were analyzed per condition) (data are mean ± s.d.; $n = 3$ biological replicates). Scale bar, 20 μm. **g** Clonogenic survival of U2OS after 10 days of siRNA treatments, as indicated. Left, representative images; right, quantification. Each bar indicates the average of observed colonies normalized to control siRNA treatment as 100%; $n = 4$ biological replicates. **h** A model depicting the MCM biogenesis pathway (i), supporting optimal origin licensing (ii), replication fork speed, and stable genomes. Misregulation of this pathway leads to reduced chromatin-bound MCMs, accelerated fork progression, and genome instability (see text for details). $P$ values were calculated by one-way ANOVA with Tukey's test (**a, g**) or Šidák's test (**b–d, f**), n.s. indicates $p > 0.1$. Source data are provided as a Source data file.

1:10,000), horse anti-mouse IgG (Cell Signaling, 7076, 1:5,000), and goat anti-rabbit IgG (Cell Signaling, 7074, 1:5,000) antibodies.

## HaloTag labeling protocol

For single-fluorescence HaloTag labeling, the U2OS or MCMBP-KO cells with Halo-tagged MCM4 and its derivatives were pulsed with JFX554-HaloTag ligand at a final labeling concentration of 50 nM for 30 min prior to fixation. For all dual HaloTag labeling, MCM4-Halo U2OS cells were incubated with JF549 HaloTag ligand (50 nM) for 30 min, followed by three washes with phosphate-buffered saline (PBS) buffer (Thermo Fisher Scientific, 14190250). The cells were then incubated with complete DMEM medium for 1 h, washed three times with PBS, and incubated in fresh DMEM with JF646 HaloTag ligand (200 nM) for 24 h. After HaloTag labeling, the cells were processed for immunofluorescence staining.

## Immunofluorescence (IF) staining

The cells were grown on round glass coverslips (12 mm diameter, #1.5, VWR, MENZCB00120RAC20). For IF staining of chromatin-bound proteins (y-axis in QIBC plots marked as "chromatin bound" protein pool), ice-cold cytoskeleton (CSK) buffer (10 mM Hepes pH 7.5; 300 mM sucrose; 100 mM NaCl; 3 mM MgCl$_2$ and 0.5% Triton X-100) was used to pre-extract the cells for 10 min at room temperature. After CSK buffer incubation, coverslips were washed three times with PBS buffer and fixed using 4% buffered formaldehyde (VWR, 9713.1000) for 20 min at room temperature. For staining the total nuclear pool of proteins (y-axis in QIBC plots marked as "nuclear pool"), 4% buffered formaldehyde was added to coverslips to fix the cells for 20 min at room temperature without pre-extraction. After fixation, the cells were washed three times with PBS and permeabilized using ice-cold PBS containing 0.2% TritonX-100 for 5 min at room temperature. Primary and secondary antibodies were diluted in fresh DMEM supplemented with 10% FBS. The coverslips were first incubated with the primary antibody cocktail for 90 min at room temperature, washed two times with PBS, and then incubated with the secondary antibody cocktail supplemented with 0.5 μg per ml 4',6-Diamidino-2-Phenylindole, Dihydrochloride (DAPI; Thermo Fisher Scientific, D1306) for 45 min at room temperature. After primary and secondary antibody staining, the coverslips were washed three times with PBS and twice with distilled water, air dried, and mounted on grease-free slides using Mowiol-based mounting medium (12% Mowiol 4-88 (Merck, 81381-250 G), 30% glycerol, 0.12 M Tris-HCl pH 8.5).

For IF staining in Fig. 5l, m and Supplementary Fig. 6e, f, samples were prepared as follows. A total of 10⁴ U2OS cells were plated in a 96-well plate (Celvis; P96-1.5H-N). Following treatment, the cells were washed with 1× PBS, fixed with 3.2% paraformaldehyde, permeabilized with 0.5% Triton X-100 in PBS for 10 min and blocked in 3% BSA in PBS for 1 h. The cells were then incubated with the indicated primary antibodies for 2 h, followed by the corresponding anti-mouse or anti-rabbit secondary antibodies for 1 h. Nuclei were visualized using DAPI

(Merck, D9542-5mg, 1:1,000). The cells were mounted with AD-Mount F (ADVI, CZ).

## Proximity ligation assay (PLA)

For PLA, wild-type U2OS cells were treated with required siRNAs (siCTRL or siDCAF12; 10 nM, 72 h) and grown on round coverslips (12 mm diameter). The cells were fixed (without pre-extraction) and permeabilized as described in the IF staining section. Cells were then blocked with DMEM containing 10 % FBS for 1 h at room temperature, followed by incubation with antibodies (diluted in 1:2000 in DMEM containing 10% FBS) for 1 h at room temperature. The coverslips were then washed three times with PBS and incubated with secondary antibody PLA probes (anti-rabbit PLUS, Sigma–Aldrich, DUO92002; anti-mouse MINUS, Sigma–Aldrich, DUO92004) for 1 h at 37 °C in a humidity chamber. After incubation, the coverslips were washed three times with PBS, and the in-situ proximity ligation (30 min at 37 °C in a humidity chamber) followed by polymerization (100 min at 37 °C in a humidity chamber) was performed using the Duolink In-Situ Detection Kit (Sigma-Aldrich, DUO92008). The nucleus was counterstained with DAPI. Finally, the coverslips were washed three times with PBS and twice with distilled water, air dried, and mounted on grease-free slides using Mowiol-based mounting medium.

## Quantitative image-based cytometry

The images for experiments (Figs. 1a–f; 2a–d, k, l; 4g–l; Supplementary Figs. 1b, c, e–h; 3e–i; 6i, j) were acquired using an inverted screening microscope ScanR (IX83, Evident), equipped with a UPLXAPO dry objective (20X, 0.8 NA), fast excitation and emission filter wheel for DAPI, FITC, Cy3, and Cy5 wavelengths, Lumencor Spectra X led fluorescence light source, digital monochromatic sCMOS ORCA-Flash 4.0 LT Plus camera, and ScanR acquisition software (Evident, v.3.4.1). The remaining experiments were acquired using an inverted screening microscope ScanR (IX81, Evident), equipped with UPLSAPO dry objective (20X, 0.75 NA), fast excitation and emission filter wheel for DAPI, FITC, Cy3, and Cy5 wavelengths, MT20 illumination system, digital monochrome Hamamatsu ORCA-R2 CCD camera, and ScanR acquisition software (Evident, v.2.7.1). During image acquisition, the laser intensity was set at 100% and exposure times were adjusted individually for each fluorophore. Image analysis was performed in ScanR analysis software (Evident, v.3.4.1). Automated dynamic background correction was applied to all images for each fluorescent channel separately, maintaining a threshold at a minimum of 5-fold pixel intensity above background levels for each channel. To identify individual nuclei as main objects, an intensity-threshold-based mask was generated based on the DAPI signal, which was then applied to analyze pixel intensities in different channels for each nucleus. The multiparameter analysis was exported as a table and further analyzed in Spotfire software (TIBCO, v.11.8.0). Within a single experiment, a similar number of cells were analyzed and compared for each condition. The mean fluorescence intensity of cytoplasmic MCMBP or MCMs was quantified using ImageJ software (FIJI, version 1.54i).

## Immunoprecipitation (IP)

The cells were collected and lysed in lysis buffer (150 mM NaCl, 50 mM Tris, pH 7.4, 0.4% Triton X-100, 2 mM $CaCl_2$, 2 mM $MgCl_2$, 1 mM EDTA, 5 mM NaF, supplemented with phosphatase and protease inhibitors) for 10 min on ice. For Strep-tagged constructs, the lysates were cleared by centrifugation and incubated with Strep-Tactin beads (IBA Lifesciences; 2-1206-025) for 2 h at 4 °C. After incubation, the beads were washed three times with lysis buffer containing Benzonase (Santa Cruz). Purified proteins were then eluted with Buffer E (IBA Lifesciences) for 5 min at room temperature. The inputs (0.5%) and eluates were analyzed by SDS-PAGE and western blotting. HA-tagged proteins were immunoprecipitated using anti-HA magnetic beads (Pierce™; Thermo Fisher Scientific). Immunoprecipitated proteins were eluted by adding 1× Bolt LDS Sample Buffer (Thermo Fisher Scientific) mixed with β-mercaptoethanol, followed by incubation at 95 °C for 5 min.

For MCM4-Halo IP from whole cell lysate, the cells were lysed in low salt lysis buffer (10 mM Tris-Cl pH 7.5, 150 mM NaCl, 0.5 mM EDTA, 0.5% NP-40; in Fig. 5b) or in high salt lysis buffer (10 mM Tris-Cl pH 7.5, 500 mM NaCl, 0.5 mM EDTA, 0.5% NP-40, in Supplementary Fig. 6a) supplemented with protease and phosphatase inhibitors (ROCHE, 4693116001 and 4906845001) and 250 U per ml of benzonase (Merck, E1014-25KU) on ice for 45 min. For MCM4-Halo IP from nuclear fraction (in Fig. 5d), the cells were fractionated; in brief, cells were incubated in hypotonic buffer (10 mM HEPES, pH 7; 5 mM NaCl, 0.3 M sucrose, 0.5% Triton-X100, supplemented with protease and phosphatase inhibitors) for 2 min on ice, followed by centrifugation at $1500 \times g$ for 5 min at 4 °C. The supernatant (cytoplasmic fraction) was discarded while the pellet was resuspended in nuclear buffer (10 mM HEPES, pH 7, 150 mM NaCl, 1 mM EDTA, 1% NP-40, supplemented with protease and phosphatase inhibitors) and incubated on ice for 10 min. The lysate was centrifuged at $16,000 \times g$ for 2 min at 4 °C, and the supernatant (nuclear fraction) was collected for IP. The clarified whole-cell or nuclear fraction lysates were then incubated with 20 µl of Halo-Trap agarose beads (Proteintech, ota-20) for 2 h at 4 °C. After incubation, the beads for low salt IP or nuclear fraction IP were washed twice with low salt lysis buffer and additionally twice with high salt lysis buffer. For MCM4-Halo IP under high salt conditions, the beads were washed four times with a high salt lysis buffer. The bound proteins were eluted by incubating the beads with 60 µl of Laemmli loading buffer containing NuPAGE LDS Sample Buffer (Thermo Fisher Scientific, NP0007) and NuPAGE Sample Reducing Agent (Thermo Fisher Scientific, NP0009) for 20 min at 95 °C. The inputs and elutes were processed by SDS-PAGE and western blotting.

## Mass spectrometry: sample preparation and data analysis

To investigate the protein interactome of DCAF12 (Fig. 1h–i), HEK293T-cells were co-transfected with one of the following combinations: StrepII-FLAG-tagged DCAF4 + HA-tagged CUL4A; StrepII-FLAG-tagged DCAF412 + HA-tagged CUL4A or StrepII–FLAG–tagged FBXL6 + HA-tagged CUL1 and cultured for 48 h. After 48 h, cells were harvested and lysed as described in the previous section. Protein complexes associated with each StrepII–FLAG–tagged construct were purified using Strep-TactinXT Superflow resin (IBA Lifesciences, Göttingen, Germany). Next, the eluates from the first purification were subjected to immunoprecipitation of HA-tagged CUL proteins (CUL4A or CUL1, depending on the construct). The eluates from both purification steps were digested with trypsin, desalted, and analyzed by LC–MS/MS at the Proteomics facility of the Biotechnology and Biomedicine Center of the Academy of Sciences and Charles University (BIOCEV) in Vestec, Czech Republic. In brief, peptide samples were analyzed by nano-LC–MS/MS using an EASY-Spray C18 analytical column (50 cm × 75 µm ID, PepMap C18, 2 µm particle size, 100 Å pore size; Thermo Scientific). The mobile phases consisted of buffer A (water with 0.1% formic acid) and buffer B (acetonitrile with 0.1% formic acid). Samples were first loaded onto a trap column (Acclaim PepMap 300 C18, 5 µm particle size, 300 Å pore size, 300 µm × 5 mm;

Thermo Scientific) at 15 µL/min for 4 min, using a loading buffer composed of water, 2% acetonitrile, and 0.1% trifluoroacetic acid. Peptides were then separated with a linear gradient of 4–35% buffer B over 60 min. Eluted peptides were ionized by electrospray and analyzed on an Orbitrap Fusion Tribrid mass spectrometer (Thermo Scientific). Survey scans of peptide precursors ($m/z$ 400–1600) were acquired in the Orbitrap at a resolution of 120,000 (at $m/z$ 200) with an AGC target of $5 \times 10^5$. Tandem MS (MS/MS) was performed by quadrupole isolation (1.5 Th), higher-energy collisional dissociation (HCD, normalized collision energy 30), and rapid scan analysis in the ion trap. The $MS^2$ AGC target was set to $1 \times 10^4$ with a maximum injection time of 35 ms. Precursors with charge states of 2–6 were selected for fragmentation. Dynamic exclusion was set to 45 s with a ± 10 ppm tolerance, and monoisotopic precursor selection was enabled. Data were acquired in top-speed mode with 2 s cycles. All the raw data were processed with MaxQuant (version 1.5.3.8) using the Andromeda search engine against the *Homo sapiens* UniProt database. Enzyme specificity was set to cleavage C-terminal to Arg and Lys, allowing cleavage at proline bonds and up to two missed cleavages. Carbamidomethylation of cysteine was selected as a fixed modification, and protein N-terminal acetylation, methionine oxidation, and serine/threonine/tyrosine phosphorylation were set as variable modifications. The false discovery rate (FDR) was set to 1% at both protein and peptide levels, with a minimum peptide length of 7 amino acids. The "match between runs" option was enabled to transfer identifications between LC–MS/MS runs (maximum retention time deviation 0.7 min). Label-free quantification was performed using MaxQuant's LFQ algorithms. Further downstream statistical analysis was conducted using Perseus (version 1.5.2.4).

To analyse DCAF12 interactome under normal conditions and after MG132 treatment (Fig. 5j), HEK293T cells were transfected with either an empty vector (EV) or StrepII–FLAG–tagged DCAF12 and cultured for 48 h. After 48 h, both groups were incubated for 6 h with either DMSO (control) or MG-132 prior to harvesting. Collected cells were then lysed as described in the previous section. The protein complexes were purified using Strep-TactinXT Superflow resin (IBA Lifesciences, Göttingen, Germany). Bound proteins were eluted with desthiobiotin, digested with trypsin, desalted, and analyzed by LC–MS/MS at the Proteomics Facility of the Biotechnology and Biomedicine Center of the Academy of Sciences and Charles University (BIOCEV, Vestec, Czech Republic). In brief, peptide samples were analyzed by nano-LC–MS/MS using an IonOpticks Aurora XT analytical column (25 cm × 75 µm ID, C18, 1.7 µm particle size, 100 Å pore size; IonOpticks). The mobile phases consisted of buffer A (water with 0.1% formic acid) and buffer B (acetonitrile with 0.1% formic acid). Samples were first loaded onto a trap column (Acclaim PepMap 300 C18, 5 µm particle size, 300 Å pore size, 300 µm × 5 mm; Thermo Scientific) at 15 µL/min for 4 min, using a loading buffer composed of water, 2% acetonitrile, and 0.1% trifluoroacetic acid. Peptides were then separated with a linear gradient of 4–35% buffer B over 60 min. Eluted peptides were ionized by electrospray and analyzed on an Orbitrap Ascend Tribrid mass spectrometer (Thermo Scientific). Survey scans of peptide precursors ($m/z$ 350–1400) were acquired in the Orbitrap at a resolution of 120,000 (at $m/z$ 200) with an AGC target of $5 \times 10^5$. Tandem MS (MS/MS) was performed by quadrupole isolation (0.7 Da), higher-energy collisional dissociation (HCD, normalized collision energy 30), and rapid scan analysis in the ion trap. The $MS^2$ AGC target was set to $1 \times 10^4$ with a maximum injection time of 35 ms. Precursors with charge states of 2–6 were selected for fragmentation. Raw data were processed with MaxQuant (version 2.3.0.0) using the Andromeda search engine against the *Homo sapiens* UniProt database. Enzyme specificity was set to cleavage C-terminal to Arg and Lys, allowing cleavage at proline bonds and up to two missed cleavages. Carbamidomethylation of cysteine was selected as a fixed modification, and protein N-terminal acetylation, methionine oxidation, and serine/threonine/tyrosine phosphorylation were set as variable

modifications. The false discovery rate (FDR) was set to 1% at both protein and peptide levels, with a minimum peptide length of 7 amino acids. The "match between runs" option was enabled to transfer identifications between LC–MS/MS runs (maximum retention time deviation 0.7 min). Label-free quantification was performed using MaxQuant's LFQ algorithms. Further downstream statistical analysis was conducted using Perseus (version 1.6.15.0).

## SDS-PAGE and Western blotting

Protein samples were separated by SDS-PAGE using NuPAGE 4–12% Bis-Tris gels and NuPAGE MES SDS Running Buffer (Thermo Fisher Scientific). Typically, 15–30 μg of total protein was loaded per well. After separation, proteins were transferred to an Amersham Hybond P 0.45 μm PVDF membrane (GE Healthcare). Membranes were blocked with 5% milk (Santa Cruz Biotechnology) in PBS supplemented with 0.1% Tween 20 (PBS-T) for 20 min and incubated with the indicated primary antibodies diluted in 3% bovine serum albumin (BSA; Applichem) in PBS-T overnight at 4 °C.

Protein samples in Fig. 5b, d, and Supplementary Fig. 6a were separated on mPAGE™ 4–12% Bis-Tris Precast Gels (Merck, MP41G10 or MP41G12) through standard SDS-PAGE protocol. The iBlot (Thermo Fisher Scientific, IB21001) system was used to transfer the separated proteins onto the nitrocellulose membrane (Thermo Fisher Scientific, IB23002X3). The membrane was blocked with PBS buffer containing 5% milk (Merck, 70166-500 G) and 0.1% Tween-20 (Merck, P1379-500ML) for 1 h at room temperature. The blocked membrane was incubated with primary antibody diluted in the blocking buffer overnight at 4 °C, followed by secondary antibody staining (diluted in blocking buffer) for 2 h at room temperature. HRP signal was then detected by ECL Select Western Blotting Detection Reagent (Merck, GERPN2235). Following the detection of the target protein, Restore PLUS Western Blot Stripping Buffer (Thermo Fischer Scientific, 46430) was used to strip the membrane and stain for other proteins.

## In vitro ubiquitination assay

The ubiquitination assay was performed as previously described[41], with minor modifications. HEK293T cells were transfected with pNSF-DCAF12 and treated 24 h later with 10 μM MG132 for 6 h. Cells were lysed on ice for 15 min in isotonic buffer containing 0.4% Triton X-100, supplemented with protease and phosphatase inhibitors (ROCHE, 4693124001 and 4906845001). Lysates were clarified, and Strep-tagged DCAF12 complexes were isolated using magnetic Strep-Tactin® resins at 4 °C for 2 h. Bound proteins were eluted using PierceTM 3× DYKDDDDK peptide (100 μg/ml) (Thermo Fisher Scientific, 815-968-0747). Affinity-purified DCAF12 (12 μL per reaction) was incubated with 15 μL of ubiquitination assay buffer containing 200 mM NaCl, 0.2 mM CaCl$_2$, 5 mM MgCl$_2$, 50 mM Tris (pH 7.5), 0.1 mg/mL BSA, 0.1% Triton X-100, 10% glycerol, 0.6 mM DTT (Sigma Aldrich, 3483-12-3), and 0.25 mM MG132. Where indicated, reactions were pre-incubated on ice for 40 min with recombinant neddylated CUL4A/RBX1 (R&D Systems™, E3-441). Ubiquitination was initiated by adding 0.1 μM UBE1 (Boston Biochem, E-305-025, LOT 05714716D), 10 ng/mL UBE2R1 (UBCH3 or CDC34) (Boston Biochem, E2-610-100, LOT 18101114B), 10 ng/mL UBCH5C (UBE2D3) (Boston Biochem, E2-627-100, LOT 04301914E) 1 μM ubiquitin aldehyde (Boston Biochem, U-201-050 LOT 11505318B), 0.25 μg/μL ubiquitin (Sigma–Aldrich), and ±1 mM ATP (Sigma–Aldrich, 34369-07-8), followed by incubation for 45 min in 37 °C. Reactions were stopped by adding 10 μL of 4× Bolt™ sample buffer (Thermo Fisher Scientific, B0008) and 5 μL of 2-mercaptoethanol, then heated at 95 °C for 5 min.

## DNA fiber assay

DNA fiber experiments were performed as previously described[30] using the following antibodies for CldU (anti-BrdU, rat, Abcam, ab6326, 1:200), IdU (anti-BrdU, mouse, Becton Dickinson, 347580,

1:200), goat anti-rat AlexaFluor 594 IgG (Thermo Fisher Scientific, A21209, 1:200), and goat anti-mouse AlexaFluor 488 IgG (Thermo Fisher Scientific, A11029, 1:200). The stained slides were mounted using Mowiol-based mounting medium (12% Mowiol 4-88 (Merck, 81381-250 G), 30% glycerol, 0.12 M Tris-HCl pH 8.5). DNA fiber images were acquired using an inverted screening microscope ScanR (IX81, Evident) equipped with UPLSAPO dry objective (20X, 0.75 NA), appropriate excitation and emission filters, and ScanR acquisition software (Evident, v.2.7.1). The lengths of CldU (red) and IdU (green) labeled tracks were measured using the ImageJ software (version 1.54i). For the calculation of replication fork speed (kb per min), the measured length (in μm) was multiplied by 2.59 (kb/μm, conversion factor)[42,43] and divided by the duration of labeling by CldU and IdU.

## Reverse transcriptase quantitative PCR (RT-qPCR)

Total RNA was isolated using a NucleoSpin RNA isolation kit (Macherey-Nagel, 740955.50). The cDNA was synthesized using a High-Capacity cDNA Reverse Transcription Kit (Thermo Fischer Scientific, 4387406) according to the manufacturer's recommendations. The RT-qPCR was performed using the Brilliant II SYBR Green qPCR Master Mix (Agilent, 600828) and primers (MCMBP primer pair #1: forward, ATGCCC TTCTGGGGGATAGT; reverse, CTATTCCGTGGGCAACCACT; MCMBP primer pair #2: forward, ACAGCCCCAGCTAATTCCAC; reverse, AGGG TCGTTCTTCCGCATTT; DCAF12 primer pair #1: forward, TGGTATCCAT GCCATCGAGC; reverse, AGAACCATCACGTGAGCCAG; DCAF12 primer pair #2: forward, GATCCACGGCAGCCATCATA; reverse, CTGCTAGTC TGGGCTTGGAC; CUL1 primer pair: forward, TTTGGGGCACCACAGA GATG; reverse, TCCAGGCCAATCTGCTTCAG; CUL2 primer pair: forward, GTCTTTCACTCCTTCGGGCT; reverse, ATTCCAACATGACCACG GCT; CUL3 primer pair: forward, CGAATCTGAGCAAAGGCACG, reverse, TCCATGGTCATCGGAAAGGC; CUL4A primer pair: forward, CAGGTA CAACCTCGAGGAGC; reverse, TGGTTTCTGTGTGCTGTGGT; CUL4B primer pair: forward, CGACCCAAAGGACGGATGAT; reverse, GCCGAA TCCCTGGGTTGTAA; CUL5 primer pair: forward, GCTAATCCGAAGG AGTCGGG; reverse, TGCTGCCCTGTTTACCCATT).

## Colony formation assay

U2OS cells were transfected with control siRNA or siRNA against DCAF12, MCMBP, or both. After 24 h of siRNA transfection, the cells were seeded onto 6-well plates (400 cells per well). After 10 days, cells were fixed with 4% buffered formaldehyde and stained with 0.1% crystal violet diluted in 20% ethanol for 30 min at room temperature. The plates were washed twice with distilled water and air-dried overnight before colony counting.

## Statistical analysis

Statistical analysis was performed using GraphPad Prism 10.1.2. Experiments were not randomized, and no blinding was used during data analysis. Sample size, statistical tests, and number of replicates for all experiments are specified in the figure legends.

## Reporting summary

Further information on research design is available in the Nature Portfolio Reporting Summary linked to this article.

## Data availability

The proteomic datasets have been deposited to the ProteomeXchange consortium via the MassIVE partner repository with the identifiers PXD055947 and PXD067954. Any additional data or information in support of this study are available from the corresponding authors upon request. Source data are provided with this paper.

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

## Acknowledgements

The research work in the Sedlackova laboratory was supported by the Czech Science Foundation Junior Star (22-20303M), the European Union's Horizon 2022 Widera Talent program (ERA grant agreement no. 101090292), EMBO Installation Grant (IG-5689-2024), and Jihomoravske centrum pro mezinarodni mobilitu (JCMM) project scholarship for foreign students. L.C. was supported by the Czech Science Foundation grant (25-18085S). K.K. was partially supported by the Charles University Grant Agency (78223). K.K., K.J., and K.O. were partially supported by Charles University Grant (SVV 260637). We acknowledge the Light Microscopy Core Facility, IMG, Prague, Czech Republic, supported by MEYS – CZ.02.1.01/0.0/0.0/18_046/0016045, MEYS – CZ.02.01.01/00/

23_015/0008205 and MEYS – LM2023050. Proteomics analyses were performed at the OMICS - MS Core Facility at BIOCEV Research Center, Faculty of Science, Charles University.

We sincerely thank Jiri Polasek, Tomas Pop, and Tomas Jendrulek for the technical maintenance of high-content imaging microscopes at IBP. We also thank Ivan Novotny for his support with the high-content imaging at IMG. We thank Kumar Somyajit and Sugith Babu Badugu for their help with PLA. We thank Luke Lavis, who generously provided JF549- and JF646-HaloTag ligands. We would like to extend our special thanks to Jiri Lukas and Michele Pagano for providing essential reagents and for their critical reading of the manuscript. Additionally, we would like to thank all members of the Sedlackova and Cermak laboratories for their stimulating discussions and insightful comments on the manuscript.

## Author contributions

H.P.S. and L.C. conceived the project and wrote the manuscript. H.P.S. lab contribution: A.K.Y. generated cell lines, performed the majority of QIBC, immunoprecipitation, analyzed data, and contributed to experimental design and concept development. S.N. performed colony formation assay, qPCR experiments, contributed to cell line generation, and antibody validation. J.K. contributed to cell line generation. H.P.S. devised the MCM biogenesis concept, designed and performed QIBC, DNA fibers, and genome instability experiments, analyzed data, prepared figures, wrote an original draft, and reviewed and edited the manuscript. L.C. lab contribution: L.C.: Conceptualization; investigation; methodology; writing—original draft; writing—review and editing; supervision; funding acquisition. A.A.: Investigation; methodology. K.O.: Investigation; methodology; visualization. K.J.: Investigation; methodology; visualization. K.K.: Investigation; methodology; visualization. N.D.: Investigation; methodology. All authors read and commented on the manuscript.

## Competing interests

The authors declare no competing interests.
