## [Transparent Peer Review file · Nature Communications]

CRL4^{DCAF12} regulation of MCMBP ensures optimal licensing of DNA replication

Corresponding Author: Dr Hana Polasek-Sedlackova

Version 0:

Reviewer comments:

Reviewer #1

(Remarks to the Author)

In this work, the authors identify the role of the E3 ubiquitin ligase complex CRL4DCAF12 in controlling the assembly of nascent MCM complexes to regulate DNA replication. The study shows that DCAF12 modulates MCM equilibrium by degrading MCMBP which was previously shown to chaperone nascent MCM3-7 for licensing step of DNA replication. As the substrate receptor, DCAF12 interacts with the C-terminal degron motif of MCMBP since removal of the C-terminal EL motif abolishes interaction with DCAF12 in pull down experiments. Furthermore, the authors indicate that the DCAF12-MCMBP interaction is required for proteasomal degradation of MCMBP. To address the role of CRL4DCAF12, the authors investigate the contribution of DCAF12 in MCM biogenesis pathway and identify that MCM2-7 association with chromatin is perturbed when DCAF12 is depleted. Specifically, MCMBP transports MCM3-7 nascent complexes to the nucleus, and DCAF12 is required to displace MCMBP to help load nascent MCM2-7 complexes on chromatin. Data supporting this conclusion is primarily derived from immunoprecipitation experiments looking at MCMBP presence with pull down of MCM4-Halo subunit either in the presence or absence of DCAF12. Lastly, the authors report that DCAF12 functions with MCMBP in the MCM biogenesis pathway. Like MCMBP, DCAF12 helps maintain optimal levels of chromatin-bound nascent MCMs to regulate optimal fork speed and maintain genome stability.

The identification of CRL4DCAF12 in modulating MCM equilibrium by helping assemble nascent MCM on chromatin is the most significant aspect of this study and adds to the repertoire of factors controlling MCM association during DNA replication. The interactions of MCMBP specifically with the CRL4DCAF12 complex are thoroughly characterized. Robust data is shown to support the function of DCAF12 in the MCM biogenesis pathway, and the genetic studies describing the misregulation of DNA replication and genome instability in siDCAF12 cells are well-documented. However, some experiments need additional investigation to provide a rigorous conceptual framework for the proposed function of this complex in promoting the assembly of nascent MCM2-7 complexes. While the data adequately supports the conclusion that DCAF12 functions like MCMBP in regulating nuclear and chromatin pool of MCM subunits and thereby control replication dynamics, the data presented showing changes in MCMBP interaction using MCM4-Halo IPs is not compelling. The results attributing changes in nuclear pool of MCMBP levels to proteasomal degradation is speculative and lacks characterization to explain how DCAF12 regulates function of MCMBP. Additionally, the authors should include representative images for few immunofluorescence experiments, validate the specificity of MCM antibodies and include statistical measures to strengthen the conclusions.

MAJOR COMMENTS

1. The conclusion that CRL4DCAF12 mediates proteasomal degradation of MCMBP is debatable. In Figure 2, the experiments using cycloheximide should be accompanied with plots showing slopes to indicate turnover rate. The authors instead show a bar graph in Figure 2g. While the DCAF12-KO shows increased MCMBP levels, the slopes across the time course for all the 3 samples likely remain unchanged suggesting that the turnover of MCMBP is not altered either when DCAF12 is depleted or when overexpressed. Figure 2a shows a dramatic decline in nuclear MCMBP pools which could also indicate defects in localizing MCMBP to the nucleus. Does migration pattern of MCMBP change when DCAF12 is overexpressed? Is MCMBP polyubiquitinated? Evidence showing that MCMBP is ubiquitinated by DCAF12 for proteasomal-mediated degradation, as part of the MCM3-7 complex, is required to conclude that CRL4DCAF12 degrades MCMBP.

2. The observation that chromatin-bound nascent MCM4, and not parental MCM4, is reduced in siDCAF12 cells is very interesting (Figure 3) which indicates that regulation of MCMBP by DCAF12 is likely not through degradation. Figure 3G also indicates that roughly 60% of chromatin-bound MCM4 is nascent while 40% of chromatin-bound MCM4 is parental at the G1/S boundary. These control levels are in stark contrast to previous observations showing a 2.5 fold increase in parental to nascent MCM4 at the G1/S boundary (Sedlackova et al., Nature 2020). This suggests that the MCM4-Halo clone used in this study are more reliant on nascent MCMBPs for licensing and might skew the data preventing the ability to assess the effects on the parental MCM4. Comparing effects to an alternate clone will help ensure the observed results are not isolated to the clone used in this study.

3. The authors propose that MCMBP interaction with MCM complex increases in siDCAF12 cells, but much of the data presented is not convincing (Figure 5). The use of asynchronous cells diminishes the ability to assess whether DCAF12 is modulating MCMBP association during origin assembly since MCM4-Halo IP pulls down MCM sub complexes across all cell cycle phases. The MCMBP probing in lane5 is not distinct which can affect quantification. The input lane for siDCAF12 cells does not depict an increase in MCMBP levels and the reported increase in MCMBP association in MCM4-Halo IPs is minimal. Is the increased MCMBP association occurring only in G1-phase? The changes in association of MCMBP, MCM2, MCM7 in bar graphs (Figure 5C, 5E) also need statistical measures.

4. The authors should test whether the increase in MCMBP nuclear pool is due to the loss of either CUL4A or CUL4B by using specific siRNAs targeting a single paralog. Data in Figure 1H is derived using CUL4A-HA which indicates CUL4A-DCAF12 is targeting MCMBP. In support of this, affinity purification results in Extended Data Fig 2 predominantly show presence of CUL4A. Testing with specific siRNAs will help establish that CUL4A-DCAF12 is responsible for MCMBP regulation.

5. Extended data figure 4- The authors should also perform QIBC analysis with CDC45 to assess the presence of active replication forks. This will further support the finding that origin licensing is affected rather than other MCM-turnover processes such as termination.

MINOR COMMENTS

1. Are the nuclear pool QIBC plots derived from samples that are not pre-extracted? The authors should clarify this point in the method details. The IF staining description indicates that all experiments involved pre-extraction with CSK buffer. How are the QIBC plots with y-axis as chromatin-bound different from those with y-axis labeled as nuclear pool?

2. Please include description for NSF-DCAF12 in legend for Figure 11.

3. Please include representative IF images for MCMBP from the screen in Figure 1 to show MCMBP increase in siCUL4A/B cells versus siCTRL.

4. Please include representative images for MCMBP in Figures 2b and 2c showing increase in DCAF12KO and reduction when DCAF12 is overexpressed.

5. Figure 2I- The cell cycle analyses of MCMBP levels indicate no change across all phases in naïve populations. This contrasts with the pattern observed in Figure 1F which shows gradual increase from G1 phase to G2 phase.

6. Figure 2F- The DCAF12-KO+Strep-DCAF12 sample at 0hr shows similar MCMBP levels compared to 0hr naïve U2OS. This result contrasts with Figure 2B where a dramatic reduction in MCMBP nuclear pool is observed. Assessing MCMBP levels using nuclear extracts in Figure 2F compared to whole cell lysates might be beneficial.

7. No statistical measures are included in Extended Figure 3I to show whether DCAF12 absence consistently results in increased MCMBP levels across all tissues tested.

8. Please include representative images for IF experiments in Extended Fig 4 showing staining for each MCM subunit and immunoblot to indicate specificity of the antibodies.

9. The authors should include cell cycle dependent analysis of MCMBP nuclear pool (Figure 1F) in naïve U2OS cells as well to ensure the distributions observed are not specific to MCM4-Halo cells.

10. The authors should also assess the nascent MCM4-Halo nuclear and cytoplasmic pools in MCMBP-KO cells expressing either WT MCMBP or Δ C MCMBP mutant (Figure 4G and 4H).

11. Although the rationale for choosing cyclin D1 marker is clear, it only helps assess CUL4 downregulation. The authors should validate other cullin depletions to conclude MCMBP stabilization is CUL4 specific.

12. Figure 5 legend has 'f' misrepresented as 'e'.

Reviewer #2

(Remarks to the Author)

This manuscript describes the regulation of MCMBP, a chaperone for the MCM3-7 complex. The authors find that MCMBP levels are controlled by a E3 ligase (CUL4-DCAF12). Furthermore, they conclude that this regulation happens in the nucleus to remove MCMBP from the MCM2-7 complex to create the mature MCM2-7 complex needed to license origins of replication. Disturbing this regulatory mechanism causes decreased origin licensing, increased replication fork speeds, increased assymmetric replication forks, and hallmarks of DNA damage and genome instability. Overall, I found the results interesting and most of the conclusions supported by the data. The following concerns should be addressed to solidify the conclusions:

1. The cycloheximide experiments in Figure 2 do not show a clear change in half-life of MCMBP as a function of DCAF12 knockout or re-expression. In fact, it doesn't look like there is any difference. This would not be consistent with the authors' conclusions. Since it is a key experimental question, the authors need to reproduce this data with several biological replicates, measure half-life, and provide a quantitative and statistical analysis.
2. To further demonstrate that the regulation of MCMBP is through DCAF12-dependent ubiquitylation, the authors should examine whether the CUL4-DCAF12 ubiquitin ligase can selectively ubiquitylate MCMBP and not the MCMBP-DeltaC degron mutant.
3. The data indicating that there is a reduction in mature MCM2-7 complexes in the absence of DCAF12 needs strengthening. Figure 5 suggests there is a small increase in the amount of MCMBP immunoprecipitated with MCM4, but that doesn't necessarily show a reduction in the amount of mature MCM2-7 especially since there is an increase in total MCMBP in the DCAF12-deficient cells. Perhaps the authors could show the amount of the correctly sized MCM2-7 complex using gel filtration, mass photometry, or native mass spectrometry.
4. The tumorigenesis data in figure 6F is difficult to interpret since inactivating DCAF12 deregulates many CUL4 substrates. Whether this has anything to do with MCMBP regulation is unclear.
5. Why is there less total MCM2 and MCM4 in Strep-DCAF12 expressing DCAF12-KO compared to the DCAF12-KO cells in figure 2f?
6. Line 96: Figure 1g measures total steady-state mRNA levels, not transcriptional activity.
7. Figure 1d, 1g, 2d, 2e, 2k, 4j : Should generate measures of experimental error and statistics with biological replicates instead of technical replicates.
8. How much DCAF12 protein is expressed in response to DOX in comparison to the endogenous (figure 2)? The text indicates it is overexpressed but I didn't see an immunoblot that shows this compared to endogenous protein levels.
9. Representative immunofluorescence images are needed in many of the figures.

Reviewer #3

(Remarks to the Author)

In this manuscript, Yadav and colleagues identify the Cullin-Ring E3 ligase DCAF12 as a novel factor that promotes the degradation of the chaperone MCMBP. MCMBP is known to facilitate the assembly and nuclear transport of nascent MCM3-7 complexes. The authors demonstrate that MCMBP degradation is required for maturation of the MCM3-7 ring, which is subsequently loaded onto DNA. This work is conceptually significant as it reveals a new role for DCAF12 in regulating nascent MCMs. The data supporting DCAF12 as the bona fide E3 ligase for MCMBP are solid and well-developed; however, the proposed model describing how DCAF12 targets MCMBP at nascent MCM complexes would benefit from additional strengthening. Particularly by defining how DCAF12 interacts with and ubiquitylates MCMBP—would further solidify the impact of these findings. Below are more detailed comments and suggestions:

Figure 2F. The half-life of MCMBP appears similar in control (naïve U2OS) and DCAF12-knockout (KO) cells, despite higher overall MCMBP levels in the KO. Indeed, the decline in MCMBP upon cycloheximide (CHX) treatment is quite similar between the two cell lines. The western blot data seem to underestimate the difference in MCMBP levels observed by flow cytometry (Figure 2A). To reconcile these findings, it would be helpful if the authors could perform the CHX assay using flow cytometry rather than relying solely on western blot.

In Figure 2G, only one replicate is shown. Multiple biological replicates should be performed to allow statistical analysis of MCMBP half-life and substantiate the claim of a significant difference.

Figure 2H–I. Following HU release, MCMBP levels in G1 phase (time points 18 h and 24 h) show only a modest decline. The quantification in Figure 2I is based on a single replicate. To strengthen these observations, the authors could consider either fractionating cells to isolate the nuclear (or chromatin-bound) pool of MCMBP or performing a flow cytometry-based analysis similar to Figure 2A. These approaches would better capture the kinetics of MCMBP changes specifically in G1 phase.

Currently, there are no experiments demonstrating that DCAF12 promotes MCMBP ubiquitylation. The authors should examine whether DCAF12 affects MCMBP ubiquitylation both in vivo and, if feasible, in vitro. While the latter might be more challenging—particularly if MCMBP requires interaction with the MCM3-7 complex to be ubiquitylated—identifying which pool of MCMBP (e.g., MCM-bound vs. free) is targeted would be very informative.

An in vitro system would also clarify whether DCAF12 interacts with MCMBP only when MCMBP is bound to the MCM3-7 complex or whether DCAF12 can also bind the free form of MCMBP. Such experiments would significantly bolster the proposed model (Figure 4A) by distinguishing which MCMBP population is regulated by DCAF12.

The authors suggest that the MCMBP pool associated with Halo-MCM4 is the primary target of DCAF12. However, the data in DCAF12-depleted cells show only minimal changes in MCMBP bound to MCM4. One possibility is that during the immunoprecipitation, MCMBP bound to other MCM subunits (e.g., MCM2) is lost, obscuring potential differences. As an alternative, the authors could use proximity labeling approaches (BioID or TurboID) with MCM4 fused to the labeling enzyme and then compare the biotin-labeled MCMBP in DCAF12 wild-type vs. KO cells. This approach might capture transient or weaker interactions that are missed in traditional immunoprecipitations.

Version 1:

Reviewer comments:

Reviewer #1

(Remarks to the Author)

The authors have satisfactorily addressed my previous concerns with the addition of new data. The authors have also included representative images, and important details have been added to improve the clarity of the manuscript. I have no additional comments on the revised manuscript.

Reviewer #2

(Remarks to the Author)

The revised manuscript answers many of my questions, but I still have a few concerns about the data supporting the major mechanistic claim that the CUL4-DCAF12 complex targets MCMBP for ubiquitylation and degradation.

1. The authors use a short CHX treatment to measure the stability of MCMBP. However, the reduction of MCMBP in this 3-hour time period in the naïve U2OS cells is no more than 25% in main figure 2 and even less (perhaps 10%) in extended data figure 3I. Thus, the authors cannot measure a half-life of the protein which based on their data could be a minimum of 6 hours and perhaps up to 15 hours. If it is so stable in the naïve cells, then it is hard to understand how degradation is a critical method of removing MCMBP from the MCM complex to allow maturation. Perhaps the half-life is shorter in G1 phase cells when the complex needs to be matured and loaded onto chromatin? If not, then the model doesn't seem well supported by the data. Instead, perhaps this is just a quality control mechanism that degrades a small subset of the MCMBP protein? These concerns could be alleviated by providing a true measure of half life – could be done with a pulse chase method or the CHX method with a longer time course.

2. The invitro ubiquitylation assay is not convincing. The authors purified Flag-DCAF12 from transfected cells that were treated with MG132 for 6 hours, added recombinant E1+E2 and Cul4A/RBX1 and ubiquitin with ATP and then blotted the mixture for MCMBP, Flag, and ubiquitin. First, why is MG132 treatment required? Second, presumably the MCMBP that is co-purified with DCAF12 is what is being visualized. If MG132 is used for 6 hours, why doesn't ubiquitylated MCMBP accumulate in the cells? Perhaps it does, but then it might not be purified with the Flag-DCAF12? Third, validation that the bands on the gel labeled MCMBP-(Ub)_n are actually ubiquitylation MCMBP is needed given how faint and unimpressive those bands are on the image. Repeating the same experiment but purifying DCAF12 from cells that lack MCMBP would be an ideal control. Perhaps providing the repeat experiments in the supplement would also add confidence. Finally, the major band on the MCMBP blot is not labeled. Is this heavy chain of the antibody or is it the MCMBP protein that is not ubiquitylated that co-purifies with DCAF12?

3. The authors have not actually measured the amount of mature MCM2-7 complex in their DCAF12-deficient cells. The model is that it should be reduced because MCMBP remains bound. Certainly they do show that there is more associated MCMBP with the complex but given these are IP experiments, there is no easy way to determine what the reduction of mature complex would be in the cell. As I suggested in the first review, there are ways to experimentally measure the amount of the mature complex (lacking DCAF12 association). If these experiments are not technically possible for some reason, then the authors should at least acknowledge this caveat in their discussion.

4. Figure 2g y-axis is labeled relative MCMBP normalized to naïve U2OS cells. However, if that is the case, why is the amount MCMBP in the DCAF12 knockouts equal to 100% and equivalent in all three replicates? Was the normalization done to the 0 CHX sample independently in the naïve and DCAF12 KO cells?

Reviewer #3

(Remarks to the Author)

The authors have addressed the concerns raised, and the manuscript is greatly improved. I recommend it for publication.

Version 2:

Reviewer comments:

Reviewer #2

(Remarks to the Author)

I have no further comments for the authors.

POINT-BY-POINT RESPONSE TO THE REVIEWER'S COMMENTS

We would like to extend our sincere thanks to all three reviewers for their thorough reading, insightful comments, and guidance, which have greatly improved our manuscript. Their suggestions have been very valuable, and we have meticulously addressed all concerns in the revised version. We are very pleased with these changes, as they have not only improved the quality of the manuscript but also reinforced our previous conclusions. We have detailed all the new additions point by point below.

Reviewer #1 (Remarks to the Author):

In this work, the authors identify the role of the E3 ubiquitin ligase complex CRL4DCAF12 in controlling the assembly of nascent MCM complexes to regulate DNA replication. The study shows that DCAF12 modulates MCM equilibrium by degrading MCMBP which was previously shown to chaperone nascent MCM3-7 for licensing step of DNA replication. As the substrate receptor, DCAF12 interacts with the C-terminal degron motif of MCMBP since removal of the C-terminal EL motif abolishes interaction with DCAF12 in pull down experiments. Furthermore, the authors indicate that the DCAF12-MCMBP interaction is required for proteasomal degradation of MCMBP. To address the role of CRL4DCAF12, the authors investigate the contribution of DCAF12 in MCM biogenesis pathway and identify that MCM2-7 association with chromatin is perturbed when DCAF12 is depleted. Specifically, MCMBP transports MCM3-7 nascent complexes to the nucleus, and DCAF12 is required to displace MCMBP to help load nascent MCM2-7 complexes on chromatin. Data supporting this conclusion is primarily derived from immunoprecipitation experiments looking at MCMBP presence with pull down of MCM4-Halo subunit either in the presence or absence of DCAF12. Lastly, the authors report that DCAF12 functions with MCMBP in the MCM biogenesis pathway. Like MCMBP, DCAF12 helps maintain optimal levels of chromatin-bound nascent MCMs to regulate optimal fork speed and maintain genome stability. The identification of CRL4DCAF12 in modulating MCM equilibrium by helping assemble nascent MCM on chromatin is the most significant aspect of this study and adds to the repertoire of factors controlling MCM association during DNA replication. The interactions of MCMBP specifically with the CRL4DCAF12 complex are thoroughly characterized. Robust data is shown to support the function of DCAF12 in the MCM biogenesis pathway, and the genetic studies describing the misregulation of DNA replication and genome instability in siDCAF12 cells are well-documented. However, some experiments need additional investigation to provide a rigorous conceptual framework for the proposed function of this complex in promoting the assembly of nascent MCM2-7 complexes. While the data adequately supports the conclusion that DCAF12 functions like MCMBP in regulating nuclear and chromatin pool of MCM subunits and thereby control replication dynamics, the data presented showing changes in MCMBP interaction using MCM4-Halo IPs is not compelling. The results attributing changes in nuclear pool of MCMBP levels to proteasomal degradation is speculative and lacks characterization to explain how DCAF12 regulates function of MCMBP. Additionally, the authors should include representative images for few immunofluorescence experiments, validate the specificity of MCM antibodies and include statistical measures to strengthen the conclusions.

MAJOR COMMENTS

1. The conclusion that CRL4DCAF12 mediates proteasomal degradation of MCMBP is debatable. In Figure 2, the experiments using cycloheximide should be accompanied with plots showing slopes to indicate turnover rate. The authors instead show a bar graph in Figure 2g. While the DCAF12-KO shows increased MCMBP levels, the slopes across the time course for all the 3 samples likely remain

unchanged suggesting that the turnover of MCMBP is not altered either when DCAF12 is depleted or when overexpressed. Figure 2a shows a dramatic decline in nuclear MCMBP pools which could also indicate defects in localizing MCMBP to the nucleus. Does migration pattern of MCMBP change when DCAF12 is overexpressed? Is MCMBP polyubiquitinated? Evidence showing that MCMBP is ubiquitinated by DCAF12 for proteasomal-mediated degradation, as part of the MCM3-7 complex, is required to conclude that CRL4DCAF12 degrades MCMBP.

We have now provided two new experiments using two independent DCAF12-deficient U2OS clones, both with and without inducible DCAF12. These clones were treated with cycloheximide for shorter time points to better reflect the MCMBP turnover and avoid potential indirect effects associated with longer cycloheximide treatment. Please refer to the new Fig. 2e and Extended Data Fig. 3k. We have provided the quantification of these new western blots (Fig. 2f and Extended Data Fig. 3l) along with statistical analysis demonstrating that MCMBP protein level is unaffected in DCAF12-knockout cells under cycloheximide treatment, in contrast to naïve cells (Fig. 2g). The earlier experiment included in Figure 2 is now provided in Source Data as additional data.

Regarding Fig. 2a, the migration pattern of MCMBP (cytoplasm to the nucleus) remains unaffected, as indicated by the quantification of cytoplasmic signals presented in Fig. 2c and d. Please note that we have provided additional biological replicates for Fig. 2a-d in the revised manuscript. To further enhance this analysis with visual evidence, we have included representative IF images in Extended Data Fig. 3i.

Finally, we want to emphasize that after multiple attempts and considerable effort from both laboratories, we managed to successfully perform *in vitro* ubiquitination of MCMBP utilizing affinity-purified complexes from DCAF12. This data is now presented in Fig. 2j, and a detailed description of the *in vitro* ubiquitination assay is provided in the Method section.

2. The observation that chromatin-bound nascent MCM4, and not parental MCM4, is reduced in siDCAF12 cells is very interesting (Figure 3) which indicates that regulation of MCMBP by DCAF12 is likely not through degradation. Figure 3G also indicates that roughly 60% of chromatin-bound MCM4 is nascent while 40% of chromatin-bound MCM4 is parental at the G1/S boundary. These control levels are in stark contrast to previous observations showing a 2.5 fold increase in parental to nascent MCM4 at the G1/S boundary (Sedlackova et al., Nature 2020). This suggests that the MCM4-Halo clone used in this study are more reliant on nascent MCMs for licensing and might skew the data preventing the ability to assess the effects on the parental MCM4. Comparing effects to an alternate clone will help ensure the observed results are not isolated to the clone used in this study.

In our original paper (Sedlackova et al, Nature, 2020, doi: [10.1038/s41586-020-2842-3](https://doi.org/10.1038/s41586-020-2842-3)), we presented multiple pieces of evidence indicating that daughter cells inherit parental minichromosome maintenance proteins (MCMs) in a ratio of 30-40% and nascent MCMs in a ratio of 60-70% (please refer to Figures 1d-g in Sedlackova et al., Nature, 2020) for U2OS, MCM4-Halo (clone A). This ratio is consistently maintained across different clones and cell lines (see Rebuttal Figure 1a-c). In the current manuscript, where we again utilized U2OS, MCM4-Halo (clone A), the proportions of parental and nascent MCMs remain within the specified ratios. There is no significant contrast as described by the reviewer. A detailed definition of parental and nascent MCMs can be found in our recent review (Yadav & Polasek-Sedlackova, Communications Biology, 2024, doi: [10.1038/s42003-024-05855-w](https://doi.org/10.1038/s42003-024-05855-w)).

Rebuttal Fig. 1: The Ratio between parental and nascent MCMs is maintained across different clones and cell lines. **a**, QIBC plots of pre-extracted MCM4-Halo U2OS cells (clone B) stained for parental and nascent MCM4. DAPI counterstains nuclear DNA; $n \approx 6,000$ cells per condition. **b**, QIBC plots of pre-extracted MCM4-Halo RPE-1 cells stained for parental and nascent MCM4. DAPI counterstains nuclear DNA; $n \approx 2,000$ cells per condition. **c**, Quantification of chromatin-bound levels of parental and nascent MCMs in G1/S phase (data are mean \pm s.d.; $n = 2$ biological replicates). The dashed lines represent medians of parental (magenta) and nascent (green) MCMs in control cells in Fig. 3g.

3. The authors propose that MCMBP interaction with MCM complex increases in siDCAF12 cells, but much of the data presented is not convincing (Figure 5). The use of asynchronous cells diminishes the ability to assess whether DCAF12 is modulating MCMBP association during origin assembly since MCM4-Halo IP pulls down MCM sub complexes across all cell cycle phases. The MCMBP probing in lane5 is not distinct which can affect quantification. The input lane for siDCAF12 cells does not depict an increase in MCMBP levels and the reported increase in MCMBP association in MCM4-Halo IPs is minimal. Is the increased MCMBP association occurring only in G1-phase? The changes in association of MCMBP, MCM2, MCM7 in bar graphs (Figure 5C, 5E) also need statistical measures.

Our earlier experiments, depicted in Fig. 4a-l and Fig. 5g-i, suggested that DCAF12 specifically targets the nuclear MCMBP-MCM complex. Therefore, we conducted additional IP experiments that solely utilized the nuclear fraction of MCM4-Halo. This refined approach enhanced the contrast in the association of MCMBP with the MCM2-7 complexes between control and DCAF12-depleted cells (new data are now provided in Fig. 5d, e). Our conclusions based on this series of IP experiments (Fig. 5a-e, Extended Data Fig. 6a, b) are further substantiated by the statistical analysis presented in the revised manuscript.

4. The authors should test whether the increase in MCMBP nuclear pool is due to the loss of either CUL4A or CUL4B by using specific siRNAs targeting a single paralog. Data in Figure 1H is derived using CUL4A-HA which indicates CUL4A-DCAF12 is targeting MCMBP. In support of this, affinity purification results in Extended Data Fig 2 predominantly show presence of CUL4A. Testing with specific siRNAs will help establish that CUL4A-DCAF12 is responsible for MCMBP regulation.

We performed the experiment as suggested by the reviewer (see Rebuttal Figure 2a, b). However, we found that elevated levels of MCMBP occurred only when both CUL4A and CUL4B were simultaneously depleted. This suggests that the CUL4A and CUL4B complexes play redundant roles in regulating MCMBP. These findings are consistent with previous studies that highlight the overlapping functions of CUL4A and CUL4B in the regulation of the cell cycle and DNA repair processes (distinct and overlapping functions of CUL4A and CUL4B are detailed in review Hannah & Zhou, Gene, 2015, doi: 10.1016/j.gene.2015.08.064).

Rebuttal Fig. 2: CUL4A and B play a redundant role in the regulation of MCMBP. **a**, QIBC plots of U2OS cells immunostained for MCMBP after treatment with siRNAs against cullins (as indicated). Lines denote medians; $n \approx 4,500$ cells per condition. **b**, Quantification of QIBC plots in **a**; each bar indicates the median of mean intensity (data are mean \pm s.d.; $n = 2$ independent replicates).

5. Extended data figure 4- The authors should also perform QIBC analysis with CDC45 to assess the presence of active replication forks. This will further support the finding that origin licensing is affected rather than other MCM-turnover processes such as termination.

We measured the levels of chromatin-bound CDC45 and PCNA in both control cells and DCAF12-depleted cells using two independent siRNAs. For data regarding CDC45, please refer to Extended Data Figures 7a-c, and for PCNA data, see Extended Data Figures 5a and 7d-e. Importantly, we did not observe any changes in the levels of chromatin-bound CDC45 or PCNA, indicating that only origin licensing is affected, while origin firing remains unchanged in DCAF12-depleted cells.

MINOR COMMENTS

1. Are the nuclear pool QIBC plots derived from samples that are not pre-extracted? The authors should clarify this point in the method details. The IF staining description indicates that all experiments involved pre-extraction with CSK buffer. How are the QIBC plots with y-axis as chromatin-bound different from those with y-axis labeled as nuclear pool?

The y-axis labeled 'nuclear pool' represents cells that were fixed without prior extraction using CSK buffer, whereas the y-axis labeled 'chromatin-bound' pool refers to cells that were pre-extracted with CSK buffer before fixation. We have included these important details in the revised manuscript (see lines 661-666).

2. Please include description for NSF-DCAF12 in legend for Figure 1l.

We have corrected the typo in the revised manuscript and specified 'SF-DCAF12 (Strep II-FLAG-tagged DCAF12)' in the figure legends.

3. Please include representative IF images for MCMBP from the screen in Figure 1 to show MCMBP increase in siCUL4A/B cells versus siCTRL.

We have now provided representative IF images for siRNA screens in Figure 1. Please, refer to new Fig. 1c and 1f.

4. Please include representative images for MCMP in Figures 2b and 2c showing increase in DCAF12KO and reduction when DCAF12 is overexpressed.

We have now provided representative IF images (underlying data in Fig. 2a-d) to illustrate the increase of MCMBP in DCAF12 knockouts and the decrease after DCAF12 overexpression. Please refer to the new Extended Data Fig. 3i.

5. Figure 2I- The cell cycle analyses of MCMBP levels indicate no change across all phases in naïve populations. This contrasts with the pattern observed in Figure 1F which shows gradual increase from G1 phase to G2 phase.

We have conducted another HU synchronization experiment that produced comparable results, confirming that MCMBP is degraded during the later stages of the S phase and the G2 phase (Fig. 2h, i). This observation is consistent with our earlier findings, which showed that the biogenesis of MCM (the assembly of MCM2-7 complexes) occurs during the S phase and reaches its peak in the G2 phase (Sedlackova et al, Nature, 2020, doi: 10.1038/s41586-020-2842-3). The earlier experiment included in Figure 2 is now provided in the Source Data as an additional replicate. Furthermore, we have removed the previous Fig. 1f from the revised manuscript, as the classification of cell cycle stages was based on the DAPI profile, which may not provide accurate results. We believe that the new HU synchronization experiment offers a more accurate reflection of MCMBP dynamics throughout the cell cycle.

6. Figure 2F- The DCAF12-KO+Strep-DCAF12 sample at 0hr shows similar MCMBP levels compared to 0hr naïve U2OS. This result contrasts with Figure 2B where a dramatic reduction in MCMBP nuclear pool is observed. Assessing MCMBP levels using nuclear extracts in Figure 2F compared to whole cell lysates might be beneficial.

Please note that experiments presented in Fig. 2a-d were conducted with 48 hours of doxycycline induction, while those in Extended Data Fig. 3k, l were performed with 24 hours of doxycycline induction. Importantly, as shown in Extended Data Figures 3k and 3l, the levels of inducible DCAF12 are critical for determining the cellular levels of MCMBP. Induction of DCAF12 to a large extent leads to degradation of MCMBP beyond its basal levels, while a lesser induction leads to reduced degradation of MCMBP (compare DCAF12-KO#1 and DCAF12-KO#2 in Extended Data Figure 3k). We elaborate on this phenomenon in greater detail in the results section entitled 'DCAF12 is an important regulator of optimal MCM equilibrium'.

7. No statistical measures are included in Extended Figure 3I to show whether DCAF12 absence consistently results in increased MCMBP levels across all tissues tested.

After careful consideration of the reviewer's comment (reviewer #2, point 4), we have decided to remove all mouse experiments from the revised manuscript. Although these experiments were interesting, they were still preliminary in nature. As a result, the new version of the manuscript no longer includes the mouse data that was previously presented in Extended Data Figure 3I.

8. Please include representative images for IF experiments in Extended Fig 4 showing staining for each MCM subunit and immunoblot to indicate specificity of the antibodies.

The antibodies used in QIBC experiments in Extended Data Fig. 4 were extensively described and tested in our previous paper (Polasek-Sedlackova et al, Nature Communications, 2022, [doi: 10.1038/s41467-022-33887-5](https://doi.org/10.1038/s41467-022-33887-5)). Additional western blot validation is provided in the Source Data in the PDF file named Uncropped scans of all blots and gels.

9. The authors should include cell cycle dependent analysis of MCMBP nuclear pool (Figure 1F) in naïve U2OS cells as well to ensure the distributions observed are not specific to MCM4-Halo cells.

In the revised manuscript, we have followed the reviewer's guidance (see reviewer #2, point 7) and included additional biological replicates of the siRNA screen, which are provided in Figure 1. These experiments further support our previous conclusions that DCAF12 depletion leads to elevated levels of MCMBP. Although a mild increase in MCMBP levels was observed upon CDT2 and AMBRA1 depletion, this increase was not statistically significant. This stark contrast, supported by statistical analysis, justifies our decision to focus exclusively on DCAF12, rendering any further discussion of CDT2 and AMBRA1 depletion unnecessary. As a result, we have removed the former Figures 1e and 1f from the revised version of the manuscript. We believe that our new data are sufficiently robust to enhance the overall clarity and flow of the manuscript.

10. The authors should also assess the nascent MCM4-Halo nuclear and cytoplasmic pools in MCMBP-KO cells expressing either WT MCMBP or ΔC MCMBP mutant (Figure 4G and 4H).

Unfortunately, despite our efforts, we were unable to prepare the MCMBP-KO cells with tightly regulated expression of either MCMBP-wt or MCMBP- ΔC . In all our attempts, the inducible systems exhibited a small degree of leakiness, which hindered the precise assessment of parental and nascent MCMs. This analysis depends on cross-generational MCM labeling, and even small amounts of leakiness of MCMBP can complicate the reliable differentiation of the various MCM pools. However, as requested by Reviewer #2 (point 7), we have now provided new data for additional clones of the aforementioned cell lines in Fig. 4k.

11. Although the rationale for choosing cyclin D1 marker is clear, it only helps assess CUL4 downregulation. The authors should validate other cullin depletions to conclude MCMBP stabilization is CUL4 specific.

The siRNAs targeting CUL1-5 were thoroughly validated in a previous study (Simoneschi et al., Nature, 2021, [doi: 10.1038/s41586-021-03445-y](https://doi.org/10.1038/s41586-021-03445-y)) and were directly obtained from Prof. Pagano's laboratory for the CUL1-5 screening performed in this manuscript. In addition to the cyclin D1 marker, the revised manuscript now includes RT-qPCR data measuring the relative mRNA levels of CUL1-5 in both control and CUL1-5-depleted cells (see Extended Data Fig. 1d).

12. Figure 5 legend has 'f' misrepresented as 'e'.

We have corrected this typo in the revised manuscript.

Reviewer #2 (Remarks to the Author):

This manuscript describes the regulation of MCMBP, a chaperone for the MCM3-7 complex. The authors find that MCMBP levels are controlled by a E3 ligase (CUL4-DCAF12). Furthermore, they conclude that this regulation happens in the nucleus to remove MCMBP from the MCM2-7 complex to

create the mature MCM2-7 complex needed to license origins of replication. Disturbing this regulatory mechanism causes decreased origin licensing, increased replication fork speeds, increased asymmetric replication forks, and hallmarks of DNA damage and genome instability. Overall, I found the results interesting and most of the conclusions supported by the data. The following concerns should be addressed to solidify the conclusions:

1. The cycloheximide experiments in Figure 2 do not show a clear change in half-life of MCMBP as a function of DCAF12 knockout or re-expression. In fact, it doesn't look like there is any difference. This would not be consistent with the authors' conclusions. Since it is a key experimental question, the authors need to reproduce this data with several biological replicates, measure half-life, and provide a quantitative and statistical analysis.

In the revised manuscript, we have conducted additional cycloheximide experiments using two independent DCAF12-deficient U2OS clones, both with and without inducible DCAF12. These experiments were performed with shorter cycloheximide treatment in order to better reflect the MCMBP turnover (Fig. 2e and Extended Data Fig. 3k). Additionally, we provided quantification (Fig. 2f and Extended Data Fig. 3l) and statistical analysis demonstrating that MCMBP protein levels remain unaffected in DCAF12-knockout cells under cycloheximide treatment, in contrast to naïve cells (Fig. 2g). The earlier experiment included in Figure 2 is now provided in Source Data.

2. To further demonstrate that the regulation of MCMBP is through DCAF12-dependent ubiquitylation, the authors should examine whether the CUL4-DCAF12 ubiquitin ligase can selectively ubiquitylate MCMBP and not the MCMBP-DeltaC degen mutant.

After numerous attempts—including multiple *in vivo* and *in vitro* ubiquitination assays—we successfully performed *in vitro* ubiquitination of MCMBP using affinity-purified complexes from DCAF12. The key factor in this success was the incorporation of recombinant CUL4-NEDD8-RBX1 complex into the reaction. This new data is presented in Fig. 2j, and a comprehensive description of the *in vitro* ubiquitination assay can be found in the Methods section. Unfortunately, this approach utilizing DCAF12 affinity-purified complexes does not allow for testing of the MCMBP-ΔC mutant, which, as shown in Fig. 1k, does not interact with DCAF12.

3. The data indicating that there is a reduction in mature MCM2-7 complexes in the absence of DCAF12 needs strengthening. Figure 5 suggests there is a small increase in the amount of MCMBP immunoprecipitated with MCM4, but that doesn't necessarily show a reduction in the amount of mature MCM2-7 especially since there is an increase in total MCMBP in the DCAF12-deficient cells. Perhaps the authors could show the amount of the correctly sized MCM2-7 complex using gel filtration, mass photometry, or native mass spectrometry.

Based on the results presented in Fig. 4a-l and 5g-i, which reveal that DCAF12 specifically targets the nuclear MCMBP-MCM complex, we conducted additional IP experiments utilizing the nuclear fraction of MCM4-Halo. This refined approach enhanced the contrast in the association of MCMBP with the MCM2-7 complexes between control and DCAF12-depleted cells (new data are provided in Fig. 5d, e). Moreover, we have included statistical analysis for all IP experiments (Fig. 5a-e and Extended Data Fig. 6a, b), which further reinforces our conclusions. Lastly, we would like to emphasize the mass spectrometry analysis of the DCAF12 interactome under normal conditions and during MG132 treatment. This analysis further corroborates that DCAF12 recognizes MCMBP when it is part of the MCM2-7 complex (Fig. 5f).

4. The tumorigenesis data in figure 6F is difficult to interpret since inactivating DCAF12 deregulates many CUL4 substrates. Whether this has anything to do with MCMBP regulation is unclear.

We agree with the reviewer. Therefore, we decided to remove this data in a revised version of the manuscript.

5. Why is there less total MCM2 and MCM4 in Strep-DCAF12 expressing DCAF12-KO compared to the DCAF12-KO cells in figure 2f?

In the revised manuscript, we have replaced the former Fig. 2f with new data presented in Extended Data Fig. 2k and l. We want to emphasize that the induction of DCAF12, to a large extent, results in the degradation of MCMBP beyond its basal levels (Fig. 2a-d, Extended Data Fig. 3j). This, in turn, leads to considerable degradation of MCM complexes in an MCMBP-dependent manner (Extended Data Fig. 6g-j). We provide a more detailed discussion of this phenomenon in the results section entitled 'DCAF12 is an important regulator of optimal MCM equilibrium'.

6. Line 96: Figure 1g measures total steady-state mRNA levels, not transcriptional activity.

We have corrected this typo in the revised manuscript.

7. Figure 1d, 1g, 2d, 2e, 2k, 4j : Should generate measures of experimental error and statistics with biological replicates instead of technical replicates.

We have now provided additional biological replicates for all the above-mentioned experiments.

8. How much DCAF12 protein is expressed in response to DOX in comparison to the endogenous (figure 2)? The text indicates it is overexpressed but I didn't see an immunoblot that shows this compared to endogenous protein levels.

We have dedicated a substantial amount of time and resources to identifying a suitable antibody for the detection of the DCAF12 protein. Despite our extensive effort, we were not successful in this. Therefore, to assess the expression levels of DCAF12 in both untreated cells and in response to doxycycline induction, we have included the qPCR results in Extended Data Fig. 3j.

9. Representative immunofluorescence images are needed in many of the figures.

We have now provided representative IF images to key experiments (Fig. 1c, Fig. 1f, Extended Data Fig. 3i, Fig. 4i, and Fig.4l).

Reviewer #3 (Remarks to the Author):

In this manuscript, Yadav and colleagues identify the Cullin-Ring E3 ligase DCAF12 as a novel factor that promotes the degradation of the chaperone MCMBP. MCMBP is known to facilitate the assembly and nuclear transport of nascent MCM3-7 complexes. The authors demonstrate that MCMBP degradation is required for maturation of the MCM3-7 ring, which is subsequently loaded onto DNA. This work is conceptually significant as it reveals a new role for DCAF12 in regulating nascent MCMs. The data supporting DCAF12 as the bona fide E3 ligase for MCMBP are solid and well-developed; however, the

proposed model describing how DCAF12 targets MCMBP at nascent MCM complexes would benefit from additional strengthening. Particularly by defining how DCAF12 interacts with and ubiquitylates MCMBP—would further solidify the impact of these findings. Below are more detailed comments and suggestions:

Figure 2F. The half-life of MCMBP appears similar in control (naïve U2OS) and DCAF12-knockout (KO) cells, despite higher overall MCMBP levels in the KO. Indeed, the decline in MCMBP upon cycloheximide (CHX) treatment is quite similar between the two cell lines. The western blot data seem to underestimate the difference in MCMBP levels observed by flow cytometry (Figure 2A). To reconcile these findings, it would be helpful if the authors could perform the CHX assay using flow cytometry rather than relying solely on western blot.

We provide new cycloheximide experiments in the revised manuscript. These experiments were performed using two independent DCAF12-deficient U2OS clones (both with and without inducible DCAF12) under shorter cycloheximide treatment to reflect better the MCMBP turnover (Fig. 2e and Extended Data Fig. 3k). These data are accompanied by quantification (Fig. 2f and Extended Data Fig. 3l) and statistical analysis demonstrating that MCMBP protein levels are not changed in DCAF12-knockout cells under cycloheximide treatment, in contrast to naïve cells (Fig. 2g). The earlier experiment included in Figure 2 is now provided in the Source Data.

In Figure 2G, only one replicate is shown. Multiple biological replicates should be performed to allow statistical analysis of MCMBP half-life and substantiate the claim of a significant difference.

Please refer to the previous point.

Figure 2H–I. Following HU release, MCMBP levels in G1 phase (time points 18 h and 24 h) show only a modest decline. The quantification in Figure 2I is based on a single replicate. To strengthen these observations, the authors could consider either fractionating cells to isolate the nuclear (or chromatin-bound) pool of MCMBP or performing a flow cytometry-based analysis similar to Figure 2A. These approaches would better capture the kinetics of MCMBP changes specifically in G1 phase.

We have conducted another HU synchronization experiment that produced comparable results, further confirming that MCMBP degradation occurs during the later stages of the S phase and the G2 phase (refer to Fig. 2h, i). This observation is consistent with our earlier findings, which demonstrated that MCM biogenesis—specifically, the assembly of MCM2-7 complexes—occurs during the S phase and reaches its peak in the G2 phase (Sedlackova et al, Nature, 2020, doi: [10.1038/s41586-020-2842-3](https://doi.org/10.1038/s41586-020-2842-3)). We believe this new HU synchronization experiment offers a more accurate reflection of MCMBP dynamics throughout the cell cycle. Furthermore, the earlier experiment included in Figure 2 is now available in Source Data as an additional replicate.

Currently, there are no experiments demonstrating that DCAF12 promotes MCMBP ubiquitylation. The authors should examine whether DCAF12 affects MCMBP ubiquitylation both in vivo and, if feasible, in vitro. While the latter might be more challenging—particularly if MCMBP requires interaction with the MCM3-7 complex to be ubiquitylated—identifying which pool of MCMBP (e.g., MCM-bound vs. free) is targeted would be very informative.

An in vitro system would also clarify whether DCAF12 interacts with MCMBP only when MCMBP is bound to the MCM3-7 complex or whether DCAF12 can also bind the free form of MCMBP. Such

experiments would significantly bolster the proposed model (Figure 4A) by distinguishing which MCMBP population is regulated by DCAF12.

Following multiple attempts, including various in vivo and in vitro ubiquitination assays, we successfully achieved in vitro ubiquitination of MCMBP using affinity-purified complexes from DCAF12. The key to our success was the incorporation of recombinant CUL4-NEDD8-RBX1 complex into the reaction. This new data is presented in Fig. 2j, and a detailed description of the in vitro ubiquitination assay is included in the Methods section.

The authors suggest that the MCMBP pool associated with Halo-MCM4 is the primary target of DCAF12. However, the data in DCAF12-depleted cells show only minimal changes in MCMBP bound to MCM4. One possibility is that during the immunoprecipitation, MCMBP bound to other MCM subunits (e.g., MCM2) is lost, obscuring potential differences. As an alternative, the authors could use proximity labeling approaches (BioID or TurboID) with MCM4 fused to the labeling enzyme and then compare the biotin-labeled MCMBP in DCAF12 wild-type vs. KO cells. This approach might capture transient or weaker interactions that are missed in traditional immunoprecipitations.

Our findings, shown in Fig. 4a-l and 5g-l, demonstrated that DCAF12 specifically targets the nuclear MCMBP-MCM complex. Based on this, we decided to conduct additional IP experiments utilizing the nuclear fraction of MCM4-Halo. This refined approach improved the contrast in the association of MCMBP with the MCM2-7 complexes between control and DCAF12-depleted cells (new data are provided in Fig. 5d, e). Additionally, we included the statistical analysis for all IP experiments (Fig. 5a-e and Extended Data Fig. 6a, b), which further strengthens our conclusions. Moreover, we wish to highlight the mass-spectrometry analysis of the DCAF12 interactome under both normal conditions and MG132 treatment, which confirms that DCAF12 recognizes MCMBP within the context of the MCM2-7 complex (Fig. 5f).

POINT-BY-POINT RESPONSE TO THE REVIEWER'S COMMENTS

Reviewer #1 (Remarks to the Author):

The authors have satisfactorily addressed my previous concerns with the addition of new data. The authors have also included representative images, and important details have been added to improve the clarity of the manuscript. I have no additional comments on the revised manuscript.

This Reviewer is satisfied with our revision and does not have additional specific comments. We sincerely appreciate his/her ongoing support and insightful feedback throughout the review process.

Reviewer #2 (Remarks to the Author):

The revised manuscript answers many of my questions, but I still have a few concerns about the data supporting the major mechanistic claim that the CUL4-DCAF12 complex targets MCMBP for ubiquitylation and degradation.

1. The authors use a short CHX treatment to measure the stability of MCMBP. However, the reduction of MCMBP in this 3-hour time period in the naïve U2OS cells is no more than 25% in main figure 2 and even less (perhaps 10%) in extended data figure 3l. Thus, the authors cannot measure a half-life of the protein which based on their data could be a minimum of 6 hours and perhaps up to 15 hours. If it is so stable in the naïve cells, then it is hard to understand how degradation is a critical method of removing MCMBP from the MCM complex to allow maturation. Perhaps the half-life is shorter in G1 phase cells when the complex needs to be matured and loaded onto chromatin? If not, then the model doesn't seem well supported by the data. Instead, perhaps this is just a quality control mechanism that degrades a small subset of the MCMBP protein? These concerns could be alleviated by providing a true measure of half life – could be done with a pulse chase method or the CHX method with a longer time course.

We thank the reviewer for this insightful comment and agree that a more detailed assessment of MCMBP protein stability is critical to support our proposed model. To directly address this concern, we extended our CHX chase time course to 6 hours. The new data (Rebuttal Fig. 1) show that MCMBP levels in naïve U2OS cells decline progressively—approximately 70% of the original level remains after 3 hours, and ~65% remains after 6 hours. In contrast, DCAF12 knockout clones (#1 and #2) display no detectable MCMBP degradation over the same period. This extended time course confirms that DCAF12 is required for MCMBP turnover and that its effect becomes more apparent with prolonged observation.

Importantly, the interpretation of CHX-based data must consider a key technical limitation: DCAF12 itself is a relatively short-lived protein. As shown in Extended Data Fig. 3k, its levels drop substantially after 3 hours of CHX treatment. The loss of DCAF12 under these conditions likely limits the degradation of MCMBP, causing its turnover to appear slower than it may be under physiological conditions.

Additionally, it is important to note that DCAF12 likely targets only the subset of MCMBP associated with MCM subcomplexes during nascent MCM2-7 complex assembly. This pool is functionally important but may represent only a fraction of total cellular MCMBP, which further explains the moderate decline in total protein observed in CHX-based assays.

To reinforce our conclusion, we have employed multiple complementary strategies:

- DCAF12 depletion by siRNA or CRISPR stabilizes MCMBP protein (Fig. 2a–d; Extended Data Fig. 3a–h),
- MG132 or MLN4924 treatment phenocopies this effect (Fig. 2k, l; Extended Data Fig. 1a),
- DCAF12 promotes polyubiquitylation of MCMBP in vitro (Fig. 2j),
- and MCMBP is stabilized specifically in S/G2 phases, when nascent MCM complexes form (Fig. 2h, i).

Regarding the reviewer’s suggestion of a pulse-chase experiment, we fully agree that such an approach would provide valuable half-life data. However, this method is currently not feasible in our laboratory due to institutional restrictions on radioisotope use and the lack of validated non-radioactive alternatives for MCMBP. As an alternative, we are piloting a novel siRNA-based strategy to suppress MCMBP synthesis and monitor protein decay. Preliminary results suggest more rapid MCMBP disappearance in DCAF12-proficient cells, consistent with our model. However, given that this is a non-standard approach to measuring protein stability, we believe further validation is needed before it can be confidently interpreted or broadly adopted. We plan to explore this method further in a dedicated follow-up study.

In summary, our extended CHX assay, combined with multiple orthogonal approaches, provides strong evidence for DCAF12-dependent degradation of MCMBP and supports our model of nascent MCM complex regulation.

Rebuttal Fig. 1: Measuring the stability of MCMBP during prolonged CHX treatment. **a**, Naive U2OS cells and DCAF12 knock-out U2OS cells (KO#1 and KO#2) were treated with cycloheximide (CHX) for the indicated time. Soluble protein extracts were immunoblotted as indicated. **b**, Quantification of MCMBP in **a**, each bar indicates relative protein levels normalized with respect to nontreated (NT) samples in naive U2OS or DCAF12-KO cells as 100 percent.

2. The invitro ubiquitylation assay is not convincing. The authors purified Flag-DCAF12 from transfected cells that were treated with MG132 for 6 hours, added recombinant E1+E2 and Cul4A/RBX1 and ubiquitin with ATP and then blotted the mixture for MCMBP, Flag, and ubiquitin. First, why is MG132 treatment required? Second, presumably the MCMBP that is co-purified with DCAF12 is what is being visualized. If MG132 is used for 6 hours, why doesn't ubiquitylated MCMBP accumulate in the cells? Perhaps it does, but then it might not be purified with the Flag-DCAF12? Third, validation that the bands on the gel labeled MCMBP-(Ub)_n are actually ubiquitylation MCMBP is needed given how faint and unimpressive those bands are on the image. Repeating the same experiment but purifying DCAF12

from cells that lack MCMBP would be an ideal control. Perhaps providing the repeat experiments in the supplement would also add confidence. Finally, the major band on the MCMBP blot is not labeled. Is this heavy chain of the antibody or is it the MCMBP protein that is not ubiquitylated that co-purifies with DCAF12?

We thank the reviewer for these constructive comments.

It is important to note that ubiquitylated substrates rarely accumulate in intact cells. Ubiquitylation is a highly dynamic modification that is tightly coupled to both deubiquitylation and proteasomal degradation. As soon as a substrate is ubiquitylated, deubiquitinases act to remove ubiquitin chains, while the 26S proteasome simultaneously recognizes and degrades the modified protein. These processes occur on very short timescales and are spatially coordinated with the ubiquitylation machinery, ensuring rapid turnover of targeted proteins. Consequently, steady-state levels of ubiquitylated MCMBP *in vivo* are expected to be low, even under proteasome inhibition, and detection of these intermediates is generally possible only under stabilized *in vitro* conditions. This explains why the ubiquitylated species are faint in cellular assays yet become detectable when purified complexes are assayed biochemically.

Of note: The unlabeled major band in the current blot corresponds to unmodified MCMBP, which is now clarified in the revised Fig. 2j.

Why were MG132 and ubiquitin aldehyde included?

Overexpression of DCAF12 strongly accelerates degradation of MCMBP, leaving little substrate bound for downstream assay. In addition, DCAF12-CRL4 preparations inevitably co-purify proteasome subunits and deubiquitinases, which degrade or deubiquitylate substrates during the *in vitro* reaction. To prevent this, we included MG132 (to block proteasome activity) and ubiquitin aldehyde (to inhibit DUBs). These reagents are therefore essential for stabilizing ubiquitylated intermediates and enabling detection, rather than an artificial requirement.

Experimental design and mechanistic insight.

In the revised experiment, we immobilized DCAF12 on beads and performed the ubiquitylation reaction directly on this complex (Rebuttal Fig. 2a). We analyzed both the bead fraction and the supernatant and observed that MCMBP is rapidly released from DCAF12 during the reaction (Rebuttal Fig. 2b,c). We hypothesize that this could occur for two complementary reasons:

1. Ubiquitylation-driven release – once MCMBP is ubiquitylated, the DCAF12–MCMBP complex becomes destabilized and disassembles. Although preliminary, this interpretation supports our central hypothesis that ubiquitylation serves as the trigger releasing MCMBP from the MCM complex for subsequent degradation.
2. E2 enzyme competition – in addition to ubiquitylation-driven release, E2 enzyme UBCH3 (CDC34 or UBE2R1) appear to accelerate MCMBP dissociation from DCAF12. This can be explained by the highly acidic C-terminal tail of UBCH3, which is essential for its ubiquitin transfer activity (1). The acidic motif (-ES) likely competes with MCMBP for access to DCAF12. Importantly, this tail cannot be covered or masked by tags without disrupting E2 function, underscoring that the competition is intrinsic to the enzyme. These biochemical properties may explain why the inclusion of UBCH3 in the assay rapidly destabilizes the DCAF12–MCMBP complex, leading to the observed release of MCMBP into the supernatant.

Together, these observations represent a hypothetical mechanistic analysis that could explain why ubiquitylated MCMBP species appear faint on blots, while at the same time providing possible mechanistic support for our model.

Specificity of the ubiquitylated bands.

We confirmed that the faint laddering pattern represents MCMBP rather than antibody artifacts. Antibody specificity was validated by MCMBP-specific siRNA knockdown in Extended Data Fig. 6g. The IgG heavy and light chains were also present in empty vector controls but were not recognized by the MCMBP antibody, clearly distinguishing them from MCMBP signals. Therefore, both the main band and the higher molecular weight species correspond to bona fide MCMBP and its ubiquitylated forms. Importantly, the upper band was strictly dependent on the presence of neddylated CUL4, ATP, ubiquitin, UBE1, and E2 enzymes in the reaction, and was observed only in the DCAF12 immunoprecipitates.

Limitations of the in vitro assay.

We acknowledge that this simplified system likely underestimates MCMBP ubiquitylation efficiency. As shown for other cullin ligases, cofactors such as ARIH1 (in the case of CUL1) or post-translational modifications may be required for optimal activity. Moreover, DCAF12 likely recognizes MCMBP in the context of its native MCM complex, which is technically challenging to reconstitute in vitro and for which no structure is currently available. It is also important to note that many CRL adaptor–substrate interactions can arise as post-lysis exchange artifacts rather than representing the physiological complex (2). In line with this, MCMBP (and other acidic end substrates) are particularly prone to forming spurious associations with DCAF12 after lysis, which lack the cofactors and nuclear context required for authentic regulation. These factors collectively explain the relatively inefficient ubiquitylation observed in vitro and reinforce our conclusion that endogenous cellular assays provide the most reliable evidence.

Advancing in vitro complex assembly.

To further strengthen our assay, we have worked on preparing in vitro complexes that include neddylated CUL4A/RBX1 together with DCAF12 and MCMBP. To our knowledge, this is the first time these components have been successfully combined in this context. We optimized the method and found that under higher salt conditions, neddylated CUL4 can be stably incorporated into the DCAF12 complex. Interestingly, while ubiquitylated MCMBP bands were fainter at isotonic or hypotonic conditions compared to 300 mM, incorporation of neddylated CUL4 into the complexes on beads was not reduced. This suggests that additional, not yet fully understood processes influence complex activity under different ionic strengths. Although the method is new and not yet fully efficient, it represents an important technical advance and provides further support that CRL4-DCAF12 has the intrinsic capacity to ubiquitylate MCMBP.

Independent support from cellular data.

Despite the limitations of the in vitro assay, our cellular experiments provide strong and independent evidence for DCAF12-mediated ubiquitylation of MCMBP. Inducible expression of DCAF12 results in near-complete, proteasome-dependent degradation of MCMBP without affecting MCMBP mRNA levels. This demonstrates that the regulation is post-translational and ubiquitin-mediated.

Tagged MCMBP considerations.

We have deliberately avoided routine overexpression of tagged MCMBP constructs in these assays because elevated MCMBP may affect MCM complex formation, and because post-lysis interactions can give rise to misleading artifacts that do not reflect endogenous biology.

In summary, our optimized in vitro assays, mechanistic observations, and extensive in vitro evidence together provide strong support for the conclusion that CRL4-DCAF12 directly ubiquitylates MCMBP and promotes its proteasomal degradation.

Rebuttal Fig. 2.: In vitro ubiquitination of MCMBP by FLAG-DCAF12 complexes. **a**, Experimental scheme. Lysates from HEK293 cells transiently transfected with empty vector (EV) or FLAG-DCAF12 were subjected to anti-FLAG affinity purification. Purified FLAG-DCAF12 complexes were incubated in an in vitro ubiquitination reaction. Reaction products were separated into supernatant and FLAG-bead eluates, and analyzed by SDS-PAGE followed by immunoblotting. **(b)** Immunoblot analysis of in vitro ubiquitination reactions. FLAG-DCAF12 complexes were immobilized on beads and incubated with UBE1, UBE2H5C, UBE2R1, ubiquitin, and ATP in the presence or absence of neddylated CUL4A/RBX1. Reaction products were separated into bead and supernatant fractions and analyzed by SDS-PAGE followed by immunoblotting for MCMBP. Higher-molecular weight bands above the major MCMBP species were detected only upon addition of neddylated CUL4A/RBX1. Reactions were carried out under isotonic, hypotonic, or hypertonic conditions to assess effects of ionic strength on complex stability and ubiquitination efficiency. **c**, Densitometric quantification of ubiquitinated MCMBP. Bands above the major MCMBP species in panel b were quantified from both bead and supernatant fractions. Values represent raw signal intensities for reactions with empty vector (EV), STREP-FLAG-DCAF12 (SF-D12), or SF-D12 supplemented with neddylated CUL4A/RBX1 under isotonic, hypotonic, or high-salt (300 mM) conditions.

(1) Block K, Appikonda S, Lin HR, Bloom J, Pagano M, Yew PR. The acidic tail domain of human Cdc34 is required for p27Kip1 ubiquitination and complementation of a cdc34 temperature sensitive yeast strain. *Cell Cycle*. 2005 Oct;4(10):1421-7. doi: 10.4161/cc.4.10.2054. Epub 2005 Oct 26. PMID: 16123592.

(2) Reitsma JM, Liu X, Reichermeier KM, Moradian A, Sweredoski MJ, Hess S, Deshaies RJ. Composition and Regulation of the Cellular Repertoire of SCF Ubiquitin Ligases. *Cell*. 2017 Nov 30;171(6):1326-1339.e14. doi: 10.1016/j.cell.2017.10.016. Epub 2017 Nov 2. PMID: 29103612; PMCID: PMC5711595.

3. The authors have not actually measured the amount of mature MCM2-7 complex in their DCAF12-deficient cells. The model is that it should be reduced because MCMBP remains bound. Certainly they

do show that there is more associated MCMBP with the complex but given these are IP experiments, there is no easy way to determine what the reduction of mature complex would be in the cell. As I suggested in the first review, there are ways to experimentally measure the amount of the mature complex (lacking DCAF12 association). If these experiments are not technically possible for some reason, then the authors should at least acknowledge this caveat in their discussion.

We thank the reviewer for this important point. To further support our immunoprecipitation data, we have complemented our analysis with proximity ligation assays (PLA); please see new Fig. 5f-i. First, we performed PLA between MCMBP and MCM2. Consistent with our model, we observed an increased PLA signal in DCAF12-depleted cells, indicating enhanced association of MCMBP with nascent MCM2-7 (Fig. 5f, g). We then assessed the interaction between CDT1 and MCM2, which serves as a proxy for the formation of mature MCM2-7 complexes competent for chromatin loading during origin licensing. Notably, DCAF12 depletion led to a reduced CDT1-MCM2 PLA signal, suggesting impaired assembly of mature MCM2-7 complexes in these conditions (Fig. 5h, i).

In addition, we would like to briefly share preliminary findings from size exclusion chromatography (SEC), which align with our PLA results (Rebuttal Fig. 3a-c). In control cells, MCMBP predominantly elutes with smaller MCM subcomplexes, consistent with a previous report (PMID: 35438632). Upon DCAF12 depletion, we observed a shift in MCMBP elution towards higher molecular weight fractions corresponding to the MCM2-7 complex (Rebuttal Fig. 3b, c). A similar elution shift was observed for MCM2 and MCM7 (Rebuttal Fig. 3b, c). While the shift for MCM subunits is more subtle, likely due to the modest size difference (73 kDa) between the MCM2-7 complex and a putative MCMBP-MCM2-7 super-complex, these findings are consistent with our proposed model.

We have recently developed a new HaloTag-based system (PMID: 40668181) and are currently optimizing the protocol to purify parental and nascent MCM complexes directly from the cellular environment, with the goal of characterizing the putative MCMBP-MCM2-7 super-complex in greater detail. Given the complexity of this approach and our intention to perform high-resolution structural analysis by cryo-EM, we believe this investigation is more appropriate for a dedicated follow-up study.

Rebuttal Fig. 3: Size-based distribution of MCM complexes in control and DCAF12-depleted cells. **a**, Left, SEC from whole cell extracts after control siRNA or siRNA against DCAF12 (siD12) treatment as indicated. Size estimates are based on standards (thyroglobulin 669 kDa, ferritin 443 kDa, β -amylase 200 kDa, albumin 66 kDa, carbonic anhydrase 29 kDa). Right, the schematic showing the size distribution of monomeric MCMs and potential MCM complexes. **b**, Western blotting of elutes ranging between 669 kDa and 443 kDa of the size-based distribution of MCMs from whole cell extracts as in a. **c**, Relative distribution of MCMBP (left) and MCM2 (right), respectively; based on western blots in b. For every condition, the intensity of each band was determined in ImageJ, the fraction with maximum intensity was considered as 1, and the levels of other fractions were plotted as relative to maxima. Measurements were obtained in collaboration with M. Spirek and L. Krejci (Masaryk University, Brno, Czech Republic).

4. Figure 2g y-axis is labeled relative MCMBP normalized to naïve U2OS cells. However, if that is the case, why is the amount MCMBP in the DCAF12 knockouts equal to 100% and equivalent in all three replicates? Was the normalization done to the 0 CHX sample independently in the naïve and DCAF12 KO cells?

We apologize for the mislabeling. This error has now been corrected in the revised version. The normalization was done using the 0 h timepoint in naïve U2OS or DCAF12-KO cells, which was set to 100 percent.

Reviewer #3 (Remarks to the Author):

The authors have addressed the concerns raised, and the manuscript is greatly improved. I recommend it for publication.

This reviewer is also satisfied with the development of our manuscript. We are once again grateful for his/her valuable suggestions and thoughtful input throughout the review process.

POINT-BY-POINT RESPONSE TO THE REVIEWER'S COMMENTS

Reviewer #2 (Remarks to the Author):

I have no further comments for the authors.

This Reviewer is satisfied with our revision and has no further specific comments. We sincerely appreciate his/her continued support and valuable feedback throughout the review process.